# INTERACTIVE LEARNING OF SINGLE-INDEX MODELS VIA STOCHASTIC GRADIENT DESCENT

**Nived Rajaraman**
Microsoft Research
nrajaraman@microsoft.com

**Yanjun Han**
New York University
yanjunhan@nyu.edu

## ABSTRACT

Stochastic gradient descent (SGD) is a cornerstone algorithm for high-dimensional optimization, renowned for its empirical successes. Recent theoretical advances have provided a deep understanding of how SGD enables feature learning in high-dimensional nonlinear models, most notably the *single-index model* with i.i.d. data. In this work, we study the sequential learning problem for single-index models, also known as generalized linear bandits or ridge bandits, where SGD is a simple and natural solution, yet its learning dynamics remain largely unexplored. We show that, similar to the optimal interactive learner, SGD undergoes a distinct "burn-in" phase before entering the "learning" phase in this setting. Moreover, with an appropriately chosen learning rate schedule, a single SGD procedure simultaneously achieves near-optimal (or best-known) sample complexity and regret guarantees across both phases, for a broad class of link functions. Our results demonstrate that SGD remains highly competitive for learning single-index models under adaptive data.

## 1 INTRODUCTION

Stochastic gradient descent (SGD) and its many variants have achieved remarkable empirical success in solving high-dimensional optimization problems in machine learning. Recent theoretical advances have provided rigorous analyses of SGD in high-dimensional, non-convex settings for a range of statistical and machine learning tasks, such as tensor decomposition (Ge et al., 2015), PCA (Wang et al., 2017), phase retrieval (Chen et al., 2019; Tan & Vershynin, 2023), to name a few. A particularly intriguing setting is that of single-index models (Dudeja & Hsu, 2018; Ben Arous et al., 2021) (and generalizations to multi-index models (Abbe et al., 2022; 2023; Damian et al., 2022; Arnaboldi et al., 2023; Bietti et al., 2025)) with Gaussian data. In this framework, each observation $(x_t, y_t)$ consists of a Gaussian feature $x_t \sim \mathcal{N}(0, I_d)$ and a noisy outcome

$$y_t = f(\langle \theta^\star, x_t \rangle) + \varepsilon_t,$$

where $f : \mathbb{R} \to \mathbb{R}$ is a known link function, $\theta^\star \in \mathbb{S}^{d-1}$ is an unknown parameter vector on the unit sphere in $\mathbb{R}^d$, and $\varepsilon_t$ denotes the unobserved noise. With a learning rate $\eta_t > 0$ and a random initialization $\theta_1 \sim \mathrm{Unif}(\mathbb{S}^{d-1})$, the SGD update for learning single-index models is given by

$$\theta_{t+1/2} = \theta_t - \eta_t(f(\langle \theta_t, x_t \rangle) - y_t)f'(\langle \theta_t, x_t \rangle) \cdot (I - \theta_t \theta_t^\top)x_t, \quad \theta_{t+1} = \frac{\theta_{t+1/2}}{\|\theta_{t+1/2}\|}. \quad (1)$$

Here, the first update is a descent step of the population loss $\theta \mapsto \frac{1}{2}\mathbb{E}\left(f(\langle \theta, x \rangle) - y\right)^2$ at $\theta = \theta_t$, whose spherical gradient[1] is estimated based on the current sample $(x_t, y_t)$. It is well known (cf. e.g. (Ben Arous et al., 2021)) that the evolution of SGD in this context exhibits two distinct phases: an initial "search" phase, during which the *correlation* $\langle \theta_t, \theta^\star \rangle$ gradually improves from $O(d^{-1/2})$ to $\Omega(1)$, followed by a "descent" phase in which the iterates $\theta_t$ converge rapidly to the global optimum $\theta^\star$, driving $\langle \theta_t, \theta^\star \rangle$ arbitrarily close to 1.

---

[1]Recall that the spherical gradient of a function $f : \mathbb{S}^{d-1} \to \mathbb{R}$ is defined as $\nabla f = Df - \frac{\partial f}{\partial r}\frac{\partial}{\partial r}$, where $Df$ is the Euclidean gradient, and $\frac{\partial}{\partial r}$ is the derivative in the radial direction.

Beyond statistical learning, single-index models have found applications in interactive decision-making problems, including bandits and reinforcement learning, where the reward is a nonlinear function of the action. An example is manipulation with object interaction, which represents one of the largest open problems in robotics (Billard & Kragic, 2019) and requires designing good sequential decision rules that can deal with sparse and non-linear reward functions and continuous action spaces (Zhu et al., 2019). This setting is known as the *generalized linear bandit* or *ridge bandit* in the bandit literature, where the mean reward satisfies $\mathbb{E}[r_t|a_t] = f(\langle\theta^\star, a_t\rangle)$ with a known link function $f$. Classical results (Filippi et al., 2010; Russo & Van Roy, 2014) show that when $0 < c_1 \leq f'(x) \leq c_2$ everywhere, both the optimal regret and the optimal learner are essentially the same as in the linear bandit case (where $f(x) = x$). Recent studies (Lattimore & Hao, 2021; Huang et al., 2021; Rajaraman et al., 2024) have considered challenging settings where $f'(x)$ could be small around $x = 0$. This line of work yields two main insights:

1. While the final "learning" phase has the same regret as linear bandits, there could be a long "burn-in" period until the learner can identify some action $a_t$ with $\langle\theta^\star, a_t\rangle = \Omega(1)$;

2. New exploration algorithms are necessary during this burn-in period, as classical methods such as UCB are provably suboptimal for minimizing the initial exploration cost.

In response to the second point, this line of research has proposed various exploration strategies for the burn-in phase that are often tailored to the specific link function $f$ and rely on noisy gradient estimates via zeroth-order optimization. In contrast, SGD offers a natural and straightforward alternative, as its intrinsic "search" and "descent" phases align well with the "burn-in" and "learning" phases encountered in interactive decision-making.

This paper is devoted to a systematic study of SGD for learning single-index models, including the aforementioned challenging setting where $f'(x)$ could be small around $x \approx 0$, within interactive decision-making settings. In these scenarios, the actions $a_t$ are no longer Gaussian, prompting us to adopt the following exploration strategy:

$$a_t = \sqrt{1 - \sigma_t^2}\theta_t + \sigma_t Z_t, \qquad Z_t \sim \text{Unif}\left(\left\{a \in \mathbb{S}^{d-1} : \langle a, \theta_t\rangle = 0\right\}\right), \qquad (2)$$

where an additional hyperparameter $\sigma_t \in [0, 1]$ governs the exploration-exploitation tradeoff. After playing the action $a_t$ and observing the reward $r_t$, we update the parameter $\theta_t$ via the same SGD as Equation (1):

$$\theta_{t+1/2} = \theta_t - \eta_t(f(\langle\theta_t, a_t\rangle) - r_t)f'(\langle\theta_t, a_t\rangle) \cdot (I - \theta_t\theta_t^\top)a_t, \quad \theta_{t+1} = \frac{\theta_{t+1/2}}{\|\theta_{t+1/2}\|}. \qquad (3)$$

By simple algebra, the stochastic gradient in Equation (3) is also an unbiased estimator of the population (spherical) gradient of $\theta \mapsto \frac{1}{2}\mathbb{E}\left(f(\langle\theta, a\rangle) - r\right)^2$ at $\theta = \theta_t$, with the distribution of $a$ given by Equation (2) and the reward $r = f(\langle\theta^\star, a\rangle) + \varepsilon$. Our main result will establish that, for a broad class of link functions, this SGD procedure, with appropriately chosen hyperparameters $(\eta_t, \sigma_t)$, achieves near-optimal performance in both the burn-in and learning phases.

**Notation.** For $x \in \mathbb{R}^d$, let $\|x\|$ be its $\ell_2$ norm. For $x, y \in \mathbb{R}^d$, let $\langle x, y\rangle$ be their inner product. Let $\mathbb{S}^{d-1}$ be the unit sphere in $\mathbb{R}^d$. Throughout this paper we will use $\theta^\star \in \mathbb{S}^{d-1}$ to denote the true parameter, $\theta_t$ to denote the current estimate, and $m_t = \langle\theta^\star, \theta_t\rangle \in [-1, 1]$ to denote the *correlation*. The standard asymptotic notations $o, O, \Omega$, etc. are used throughout the paper, and we also use $\widetilde{O}, \widetilde{\Omega}$, etc. to denote the respective meanings with hidden poly-logarithmic factors.

## 1.1 MAIN RESULTS

First we give a formal formulation of the single-index model in the interactive setting. Let $\theta^\star \in \mathbb{S}^{d-1}$ be an unknown parameter vector, and $\mathcal{A} = \mathbb{S}^{d-1}$ be the action space. Upon choosing an action $a_t \in \mathcal{A}$, the learner receives a reward $r_t = f(\langle\theta^\star, a_t\rangle) + \varepsilon_t$ for a known link function $f : [-1, 1] \to \mathbb{R}$ and an unobserved noise $\varepsilon_t$ which is assumed to be zero-mean and 1-subGaussian.

**Remark 1.** *The scaling considered here differs crucially from the prior study on learning single-index models under non-interactive environments (such as (Dudeja & Hsu, 2018; Ben Arous et al., 2021) with Gaussian or i.i.d. features). In the non-interactive setting, it is usually assumed that $x_t \sim \mathcal{N}(0, I_d)$, so that $\|x_t\| \asymp \sqrt{d}$. In the interactive setting, we stick to the convention that actions*

*belong to the unit $\ell_2$ ball, in line with settings considered in the bandit literature (Filippi et al., 2010; Russo & Van Roy, 2014; Rajaraman et al., 2024). As a consequence, sample complexity comparisons between the interactive and non-interactive settings must be made with care. We discuss this in more detail in Section 5 and compare with results established for online SGD with Gaussian features (Ben Arous et al., 2021) after normalizing for the difference in scaling.*

Throughout the paper we make the following mild assumptions on the link function $f$.

**Assumption 1.** *The following conditions hold for the link function $f$:*

1. *(monotonicity) $f : [-1, 1] \to [-1, 1]$ is non-decreasing, with $\|f\|_\infty \leq 1$;*

2. *(locally linear near $x = 1$) $0 < \gamma_1 \leq f'(x) \leq \gamma_2$ for all $x \in [1 - \gamma_0, 1]$, with absolute constants $\gamma_0, \gamma_1, \gamma_2 > 0$. Without loss of generality we assume that $\gamma_0 \leq 0.1$.*

In Assumption 1, the monotonicity condition is taken from (Rajaraman et al., 2024) to ensure that reward maximization is aligned with parameter estimation, where improving the alignment $\langle \theta^\star, a_t \rangle$ directly increases the learner's reward. In addition, when it comes to SGD, we will show in Section 5 that the population loss associated with the SGD dynamics in Equation (3) is decreasing in the correlation $m_t = \langle \theta^\star, \theta_t \rangle$ only if $f$ is increasing. Without monotonicity, there also exists a counterexample where the SGD can never make meaningful progress (cf. Proposition 1). Similar to (Rajaraman et al., 2024), this condition can be generalized to $f$ being even and non-decreasing on $[0, 1]$, which covers, for example, $f(x) = |x|^p$ for all $p > 0$. The second condition in Assumption 1 is very mild, satisfied by many natural functions, and ensures that the problem locally resembles a linear bandit near the global optimum $a_t \approx \theta^\star$. Finally, we emphasize that this local linearity does not exclude the nontrivial scenario where $f'(x)$ is very small when $x \approx 0$.

Our first result is the SGD dynamics in the learning phase, under Assumption 1.

**Theorem 1** (Learning Phase). *Let $\varepsilon, \delta > 0$. Under Assumption 1, let $(a_t, \theta_t)_{t \geq 1}$ be given by the SGD evolution in Equation (2) and Equation (3), with an initialization $\theta_1$ such that $\langle \theta_1, \theta^\star \rangle \geq 1 - \gamma_0/4$.*

1. *(Pure exploration) By choosing $\eta_t = \widetilde{\Theta}(\frac{d}{t} \wedge \frac{1}{d})$ and $\sigma_t^2 = \Theta(1)$, it holds that $m_T \geq 1 - \varepsilon$ with probability at least $1 - \delta T$, with $T = \widetilde{O}(\frac{d^2}{\varepsilon})$.*

2. *(Regret minimization) By choosing $\eta_t = \widetilde{\Theta}(\frac{1}{\sqrt{t}} \wedge \frac{1}{d})$ and $\sigma_t^2 = \widetilde{\Theta}(\frac{d}{\sqrt{t}} \wedge 1)$, with probability at least $1 - \delta T$ it holds that $\sum_{t=1}^{T} (f(1) - f(m_t)) = \widetilde{O}(d\sqrt{T})$.*

Both upper bounds in Theorem 1 are near-optimal and match the lower bounds $\Omega(\frac{d^2}{\varepsilon})$ and $\Omega(d\sqrt{T})$ for the respective tasks, shown in Theorem 1.7 of (Rajaraman et al., 2024). In other words, SGD with proper learing rate and exploration schedules achieves an optimal learning performance in the learning phase, given a "warm start" $\theta_1$ with $\langle \theta_1, \theta^\star \rangle \geq 1 - \gamma_0/2$. To search for this "warm start" through the burn-in phase, we additionally make *one* of the following assumptions.

**Assumption 2.** *There is an absolute constant $c_0 > 0$ such that $f'(x) \geq c_0$ for all $x \in [0, 1]$.*

**Assumption 3.** *The link function $f$ is convex on $[0, 1]$.*

Specifically, Assumption 2 and 3 cover two different regimes of generalized linear bandits: Assumption 2 corresponds to the classical "linear bandit" regime studied in (Filippi et al., 2010; Russo & Van Roy, 2014), and Assumption 3 covers the case with a long burn-in period where $f'(x)$ is small at the beginning, e.g. in (Lattimore & Hao, 2021; Huang et al., 2021). We will discuss the challenges in dropping the convexity assumption for the SGD analysis in Section 5.

Under Assumption 2 or 3, our next result characterizes the SGD dynamics in the burn-in phase.

**Theorem 2** (Burn-in Phase). *Let $\delta > 0$, and Assumption 1 hold. Let $(a_t, \theta_t)_{t \geq 1}$ be given by the SGD evolution in Equation (2) and Equation (3), with an initialization $\theta_1$ such that $\langle \theta_1, \theta^\star \rangle \geq \frac{1}{\sqrt{d}}$.*

1. *Under Assumption 2, by choosing $\eta_t = \widetilde{\Theta}(\frac{1}{d^2})$ and $\sigma_t^2 = \Theta(1)$, it holds that $m_T \geq 1 - \gamma_0/4$ with probability at least $1 - \delta T$, where $T = \widetilde{O}(d^2)$.*

2. *Under Assumption 3, by choosing an appropriate learning rate schedule (cf. Lemma 8) and $\sigma_t^2 = \Theta(1)$, it holds that $m_T \geq 1 - \gamma_0/4$ with probability at least $1 - \delta T$, where*

$$T = \widetilde{O}\Big(d^2 \int_{1/(2\sqrt{d})}^{1 - \gamma_0/4} \frac{m}{f'(m)^2} \mathrm{d}m\Big).$$

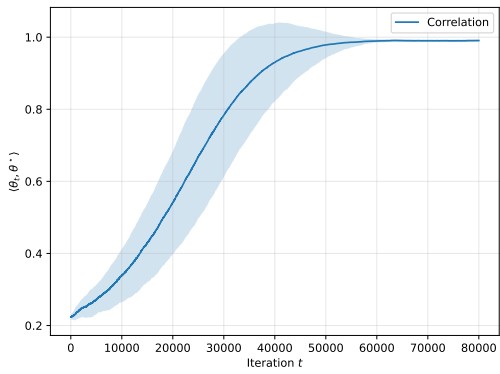 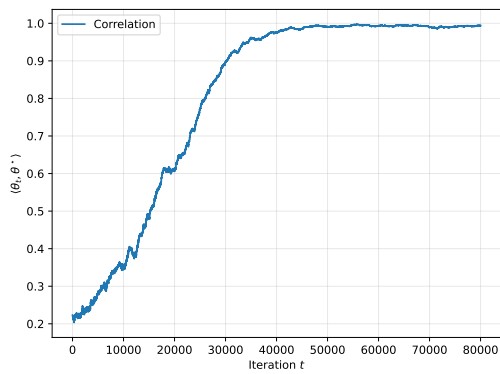

(a) Evolution of $m_t = \langle \theta^\star, \theta_t \rangle$ averaged across 100 runs. The shaded region plots one standard deviation.

(b) A single trajectory of $m_t = \langle \theta^\star, \theta_t \rangle$.

Figure 1: Correlation $m_t = \langle \theta_t, \theta^\star \rangle$ plotted as a function of $t$ in $d = 20$ dimensions, for the cubic link $f(x) = x^3$. We run interactive SGD with a constant learning rate $\eta_t = 0.002$ for all $t$, using an exploration schedule with $\sigma_t = 0.5$ until $m_t$ reaches 0.7 and $\sigma_t = 0.2$ thereafter.

Note that for $\theta_1 \sim \text{Unif}(\mathbb{S}^{d-1})$, the condition $\langle \theta^\star, \theta_1 \rangle \geq 1/\sqrt{d}$ is fulfilled with a constant probability. A simple hypothesis testing subroutine in (Rajaraman et al., 2024, Lemma 3.1) could further certify it using $\widetilde{O}((f(1/\sqrt{d}) - f(0))^{-2})$ samples. Therefore, combining Theorem 1 and 2, we have the following corollary on the overall complexity of SGD.

**Corollary 1** (Overall sample complexity and regret). *Under Assumption 1 and Assumption 2 or 3, the SGD evolution in Equation* (2) *and Equation* (3) *with proper* $(\eta_t, \sigma_t)_{t\geq 1}$ *and a hypothesis testing subroutine for initialization satisfies the following:*

1. *(Pure exploration) For* $\varepsilon, \delta > 0$, $m_T \geq 1 - \varepsilon$ *with probability at least* $1 - \delta T$, *where*

$$T = \widetilde{O}\Big(d^2 \int_{1/(2\sqrt{d})}^{1-\gamma_0/4} \frac{m}{f'(m)^2} \mathrm{d}m + \frac{d^2}{\varepsilon}\Big).$$

2. *(Regret minimization) For* $\delta > 0$, *with probability at least* $1 - \delta T$, *the regret satisfies*

$$\sum_{t=1}^{T}(f(1) - f(m_t)) = \widetilde{O}\Big( \min\Big\{T, d^2 \int_{1/(2\sqrt{d})}^{1-\gamma_0/2} \frac{m}{f'(m)^2} \mathrm{d}m + d\sqrt{T}\Big\}\Big).$$

Under Assumption 2, Corollary 1 yields an overall sample complexity bound $\widetilde{O}(d^2/\varepsilon)$ and a regret bound $\widetilde{O}(\min\{T, d\sqrt{T}\})$, both of which are known to be near-optimal, e.g., in the case of linear bandits (Lattimore & Szepesvári, 2020; Wagenmaker et al., 2022). Under Assumption 3, the upper bounds in Corollary 1 also match the best known guarantees in (Rajaraman et al., 2024), using a different algorithm based on successive hypothesis testing. In the special case $f(x) = x^p$ with odd $p \geq 3$, SGD achieves a regret bound $\widetilde{O}(\min\{T, d^p + d\sqrt{T}\})$, which is near-optimal (Huang et al., 2021; Rajaraman et al., 2024). By contrast, many other approaches, including all non-interactive algorithms (in particular, non-interactive SGD) and UCB-based methods, provably incur a larger burn-in cost of $\widetilde{\Omega}(d^{p+1})$ (Rajaraman et al., 2024). Therefore, it is striking that SGD attains optimal performance even in the burn-in phase, while simultaneously staying optimal in the learning phase. Taken together, these results highlight SGD as a natural, efficient, and highly competitive algorithm with near-optimal statistical guarantees for learning single-index models in the interactive setting. As a numerical example, Figure 1 illustrates the evolution of $m_t$ obtained by SGD with a constant learning rate for the cubic link $f(x) = x^3$, shown either as an average over 100 runs (left panel) or as a single trajectory (right panel). We observe that, despite the non-convex loss landscape and potentially non-monotonic progress, SGD consistently delivers strong performance during both the burn-in and learning phases.

## 1.2 RELATED WORK

**Single-index models.** Analyzing feature learning in non-linear functions of low-dimensional features has a long history. The approximation and statistical aspects are well understood in (Barron,

2002; Bach, 2017); by contrast, the computational aspects remain more challenging, and positive results typically require additional assumptions on the link function and/or the data distribution. Focusing on the link function $f$, a rich line of work (Kalai & Sastry, 2009; Shalev-Shwartz et al., 2010; Kakade et al., 2011; Soltanolkotabi, 2017; Frei et al., 2020; Yehudai & Ohad, 2020; Wu, 2022) has exploited its monotonicity or invertibility to obtain efficient learning guarantees under broad distributional assumptions. At the other end of the spectrum, the seminal works (Dudeja & Hsu, 2018; Ben Arous et al., 2021) developed a harmonic-analysis framework for studying SGD under Gaussian data, sparking extensive follow-up research (Abbe et al., 2022; Bietti et al., 2022; Damian et al., 2022; Ben Arous et al., 2022; Abbe et al., 2023; Zweig et al., 2023; Damian et al., 2024; Ben Arous et al., 2024; Bietti et al., 2025).

A representative finding for the single-index model is that the sample complexity of SGD is governed by the *information exponent* of the link function, i.e., the index of its first non-zero Hermite coefficient. In the interactive setting, however, where the data distribution is no longer i.i.d., the information exponent ceases to be an informative measure of SGD's performance. We defer more discussions to Section 5.

**Generalized linear bandits.** The most canonical examples of generalized linear bandits are linear bandits (Dani et al., 2008; Rusmevichientong & Tsitsiklis, 2010; Chu et al., 2011) and ridge bandits with $0 < c_1 \leq |f'(\cdot)| \leq c_2$ everywhere. In both cases, the minimax regret is $\widetilde{\Theta}(d\sqrt{T})$ (Filippi et al., 2010; Abbasi-Yadkori et al., 2011; Russo & Van Roy, 2014), attained by algorithms such as LinUCB and information-directed sampling. For more challenging convex link functions, the special cases $f(x) = x^2$ and $f(x) = x^p$ with $p \geq 2$ were analyzed in (Lattimore & Hao, 2021; Huang et al., 2021), using either successive searching algorithms or noisy power methods. These results were substantially generalized by (Rajaraman et al., 2024), which identified the existence of a general burn-in period and established tight upper and lower bounds on the optimal burn-in cost via differential equations. In particular, their upper bound strengthens Corollary 1 in the absence of convexity, using an refined algorithm of (Lattimore & Hao, 2021) during the burn-in phase and an ETC (explore-then-commit) algorithm in the learning phase. By contrast, we show that a single, much simpler SGD algorithm achieves the same upper bound for convex $f$.

A related line of work (Fan et al., 2023; Kang et al., 2025) studied the single-index model with an unknown link function, where the central idea is to estimate the score function. Their resulting algorithms are of the ETC type, and the regret guarantees rely on a positive lower bound for $f'$.

**Gradient descent in online learning and bandits.** Gradient and mirror descent are classical algorithms in online settings (including online learning and online convex optimization (Cesa-Bianchi & Lugosi, 2006; Hazan et al., 2016; Orabona, 2019)), as well as in bandit problems with gradient estimation, such as EXP3 for adversarial multi-armed bandits and FTRL for adversarial linear bandits (Lattimore & Szepesvári, 2020). A distinct feature of our work is that our SGD remains a first-order method even in this bandit problem, in contrast to the zeroth-order stochastic optimization usually used for single-index models such as (Huang et al., 2021). Moreover, the SGD dynamics for single-index models demand a more fine-grained analysis than that required by standard online learning guarantees. Further details are provided in Section 5.

### 1.3 ORGANIZATION

The rest of this paper is organized as follows. In Section 2 we present a general analysis of the SGD update, including bounds on the mean drift, stochastic term, and normalization error. In Section 3 and 4, we analyze the learning and burn-in phases, respectively. Additional discussion is provided in Section 5, and detailed proofs are deferred to the appendix.

## 2 ANALYSIS OF THE SGD UPDATE

To establish our main results Theorem 1 and 2, we first understand the properties of each SGD update in Equation (2) and Equation (3). At each time step $t$, the improvement on the correlation from $m_t := \langle \theta^\star, \theta_t \rangle$ to $m_{t+1} := \langle \theta^\star, \theta_{t+1} \rangle$ consists of three parts:

1. *Drift*: the mean improvement $\mathbb{E}[m_{t+1/2}|\mathcal{F}_t] - m_t$ of the descent step in Equation (3), where $m_{t+1/2} := \langle \theta^\star, \theta_{t+1/2} \rangle$, and $\mathcal{F}_t$ denotes all historic observations up to the end of time $t$.

2. *Martingale difference*: the stochastic term $m_{t+1/2} - \mathbb{E}[m_{t+1/2}|\mathcal{F}_t]$ with zero mean.

3. *Normalization error*: the difference $m_{t+1} - m_{t+1/2}$ due to the normalization step in Equation (3).

We will present generic lemmas to bound each of the above terms in this section, and use them to analyze the learning and burn-in phases in the next two sections. We start from the drift.

**Lemma 1** (Drift). *Let $d \geq 3$. The following identity holds for the drift:*

$$
\mathbb{E}[m_{t+1/2}|\mathcal{F}_t] - m_t
$$
$$
= \frac{\eta_t \sigma_t^2}{d-2} f'\left(\sqrt{1-\sigma_t^2}\right)(1-m_t^2) \cdot \mathbb{E}\left[f'\left(\sqrt{1-\sigma_t^2}\, m_t + \sigma_t \sqrt{1-m_t^2}\, X\right)(1-X^2)\middle|\mathcal{F}_t\right].
$$

*where $X$ follows the one-dimensional marginal of the uniform distribution over $\mathbb{S}^{d-2}$. In particular, if $m_t > 0$,*

$$
\mathbb{E}[m_{t+1/2}|\mathcal{F}_t] - m_t \geq c_{dr} \begin{cases} \frac{\eta_t \sigma_t^2}{d} f'\left(\sqrt{1-\sigma_t^2}\right)(1-m_t^2) & \text{under Assumption 2} \\ \frac{\eta_t \sigma_t^2}{d} f'\left(\sqrt{1-\sigma_t^2}\right)(1-m_t^2) f'\left(\sqrt{1-\sigma_t^2}\, m_t\right) & \text{under Assumption 3} \end{cases},
$$

*for a universal constant $c_{dr} > 0$.*

The next result concerns the subexponential concentration property of the martingale difference.

**Lemma 2** (Martingale difference). *The $\Psi_1$-Orlicz norm (i.e., the subexponential norm) of the martingale difference has the following upper bound, conditioned on $\mathcal{F}_t$:*

$$
\|m_{t+1/2} - \mathbb{E}[m_{t+1/2}|\mathcal{F}_t]\|_{\Psi_1} \leq K_t \triangleq C_{se}\sqrt{\frac{1-m_t^2}{d}}\eta_t \sigma_t f'\left(\sqrt{1-\sigma_t^2}\right),
$$

*where $C_{se} > 0$ is a universal constant.*

Based on Lemma 2, we proceed to consider the (discounted) sum of martingale differences. For $t_0 \geq 0$ and $\beta > 0$, let

$$
S_t^{t_0,\beta} := \sum_{s=t_0}^{t-1} \beta^{s-t_0}\left(m_{s+1/2} - \mathbb{E}[m_{s+1/2}|\mathcal{F}_s]\right)
$$

be a martingale adapted to $\{\mathcal{F}_t\}_{t \geq t_0}$, and $V_t^{t_0,\beta} := \sum_{s=t_0}^{t-1} \beta^{2(s-t_0)}K_s^2$ be a proxy for its predictable quadratic variation. The following result is a self-normalized concentration inequality for such processes established in (Whitehouse et al., 2023, Theorem 3.1):

**Lemma 3** (Sum of martingale differences). *Let $\eta_t \leq (C_{se}\gamma_2)^{-1}$ and $\sigma_t^2 \leq \gamma_0$ for all $t \geq 1$, and $\beta \geq 0$. For $\delta > 0$, it holds that*

$$
\mathbb{P}\left(\exists t \geq t_0 : |S_t^{t_0,\beta}| \geq C_{mt}\sqrt{V_t^{t_0,\beta} \vee 1}\log\left(\frac{1+\log(V_t^{t_0,\beta} \vee 1)}{\delta}\right) \text{ and } \beta^{t-t_0} \leq 2 \,\middle|\, \mathcal{F}_{t_0}\right) \leq \delta,
$$

*for some universal constant $C_{mt} > 0$.*

Note that when $\beta \in [0,1]$, the condition $\beta^{t-t_0} \leq 2$ is vacuous. For $\beta > 1$, this condition results in a smaller range of $t \in [t_0, t_0 + \log_\beta 2]$. Finally, we bound the normalization error $m_{t+1} - m_{t+1/2}$.

**Lemma 4** (Normalization error). *With probability at least $1 - \delta$, it holds that*

$$
m_{t+1} \geq m_{t+1/2} - C_{nm} \cdot \eta_t^2 \sigma_t^2 \left(f'\left(\sqrt{1-\sigma_t^2}\right)\right)^2 \log(1/\delta),
$$

*for some universal constant $C_{nm} > 0$. In addition, if $m_{t+1/2} \geq 0$, then with probability $1 - \delta$,*

$$
m_{t+1/2} \geq m_{t+1} \geq m_{t+1/2}\left(1 - C_{nm} \cdot \eta_t^2 \sigma_t^2 \left(f'\left(\sqrt{1-\sigma_t^2}\right)\right)^2 \log(1/\delta)\right).
$$

## 3  ANALYSIS OF THE LEARNING PHASE

In this section we analyze the SGD dynamics in the learning phase, given a "warm start" $\theta_1$ with $m_1 = \langle \theta^\star, \theta_1 \rangle \geq 1 - \gamma_0/4$.

### 3.1 PURE EXPLORATION

The crux of the proof of Theorem 1 lies in the following lemma, which shows that starting from a correlation $m_t \geq 1 - \varepsilon$, SGD will improve it to $1 - \varepsilon/2$ after $\widetilde{O}(d^2/\varepsilon)$ steps.

**Lemma 5** (Local improvement for pure exploration). *Suppose $m_t \geq 1 - \varepsilon$ for some $\varepsilon \leq \gamma_0/4$. Let $\iota := \log^2(d/\varepsilon\delta)$, and for $s \geq t$, set*

$$\eta_s \equiv \eta := \frac{c\varepsilon}{d\iota}, \quad \sigma_s^2 \equiv \sigma^2 := \gamma_0,$$

*where $c > 0$ is a small absolute constant. Then for $\Delta := Cd/\eta$ and a large absolute constant $C > 0$ independent of c, we have $m_{t+\Delta} \geq 1 - \varepsilon/2$ with probability at least $1 - \Delta\delta$.*

We call the time interval $[t, t + \Delta]$ an "epoch", and choose the learning rate based on the epoch. Lemma 5 shows that, as long as the correlation is large at the beginning of an epoch, then it must be improved in a linear rate at the end of the epoch. Therefore, by induction and a geometric series calculation, it is clear that the learning rate schedule given by Lemma 5 corresponds to $\eta_t = \widetilde{\Theta}(\frac{d}{t} \wedge \frac{1}{d})$, and Lemma 5 gives an overall sample complexity $\widetilde{O}(\frac{d^2}{\varepsilon})$ for pure exploration.

In the sequel we prove Lemma 5. We first show that by induction on $s$ that with probability at least $1 - \Delta\delta/3$, $m_s \geq 1 - 2\varepsilon$ for all $s \in [t, t + \Delta]$. The base case $s = t$ is ensured by the assumption $m_t \geq 1 - \varepsilon$. For the inductive step, suppose $m_t, \ldots, m_{s-1} \geq 1 - 2\varepsilon$. Then

$$m_s - m_t = \sum_{r=t}^{s-1} \left[ \underbrace{\left(\mathbb{E}[m_{r+1/2}|\mathcal{F}_r] - m_r\right)}_{\geq 0 \text{ by Lemma 1}} + \underbrace{\left(m_{r+1/2} - \mathbb{E}[m_{r+1/2}|\mathcal{F}_r]\right)}_{=:A_r} + \underbrace{\left(m_{r+1} - m_{r+1/2}\right)}_{=:B_r} \right].$$

Thanks to the inductive hypothesis, $K_r = O(\eta\sqrt{\frac{\varepsilon}{d}})$ for all $r \in [t, s-1]$ in Lemma 2, so Lemma 3 (with $t_0 = t, \beta = 1$) gives $|\sum_{r=t}^{s-1} A_r| = O(\eta\sqrt{\frac{\Delta\varepsilon}{d}}\log(\frac{\Delta}{\delta})) = O(\sqrt{\eta\varepsilon}\log(\frac{\Delta}{\delta})) < \frac{\varepsilon}{8}$ with probability $1 - \frac{\delta}{6}$, by choosing $c > 0$ small enough. Similarly, $\sum_{r=t}^{s-1}|B_r| = O(\Delta\eta^2\log(\frac{\Delta}{\delta})) = O(d\eta\log(\frac{\Delta}{\delta})) < \frac{\varepsilon}{8}$ with probability $1 - \frac{\delta}{6}$, by Lemma 4 and choosing $c > 0$ small enough. This implies that $m_s \geq m_t - \frac{\varepsilon}{4} > 1 - 2\varepsilon$ with probability $1 - \frac{\delta}{3}$, completing the induction.

Conditioned on the event $m_s \geq 1 - 2\varepsilon$ for all $s \in [t, t + \Delta]$, we distinguish into two regimes in this epoch. Let $T_0 \geq t$ be the stopping time when $m_s > 1 - \varepsilon/4$ for the first time.

**Regime I:** $t \leq s < T_0$. In this regime $m_s \in [1 - 2\varepsilon, 1 - \varepsilon/4]$. We show that $T_0 \leq t + \Delta$ with probability $1 - \Delta\delta/3$. If $T_0 > t + \Delta$, using the same high-probability bounds, we have

$$m_{t+\Delta} - m_t \geq \sum_{s=t}^{t+\Delta-1} \left(\mathbb{E}[m_{s+1/2}|\mathcal{F}_s] - m_s\right) - \frac{\varepsilon}{4}$$

with probability $1 - \Delta\delta/3$. By Lemma 1 with $1 - m_s^2 = \Omega(\varepsilon)$ and $\sqrt{1 - \sigma_s^2}m_s \geq 1 - \gamma_0$ for $s < T_0$, the total drift is $\Omega(\frac{\Delta\eta\varepsilon}{d}) = \Omega(C\varepsilon)$. Therefore, for a large absolute constant $C > 0$, we would have $m_{t+\Delta} \geq 1 - \varepsilon/4$, a contradiction to the assumption $T_0 > t + \Delta$.

**Regime II:** $s \geq T_0$. As shown above, this regime is non-empty with high probability. The same induction starting from $s = T_0$ shows that, with probability $1 - \Delta\delta/3$, $m_s \geq m_{T_0} - \varepsilon/4$ holds for all $s \in [T_0, t + \Delta]$. In particular, choosing $s = t + \Delta$ gives the desired result $m_{t+\Delta} \geq 1 - \varepsilon/2$.

An illustration of the above proof technique is displayed in Figure 2, shown in Appendix A.

### 3.2 REGRET MINIMIZATION

The proof of Theorem 1 for regret minimization follows similarly from the following lemma.

**Lemma 6** (Local improvement for regret minimization). *Suppose $m_t \geq 1 - \varepsilon$ for some $\varepsilon \leq \gamma_0/4$. Let $\iota := \log^2(d/\varepsilon\delta)$, $c > 0$ be a small absolute constant, and for $s \geq t$, set*

$$\eta_s \equiv \eta := \frac{c\varepsilon}{d\iota}, \quad \sigma_s^2 \equiv \sigma^2 := \varepsilon.$$

*Then for $\Delta := Cd/(\eta\varepsilon)$ and a large absolute constant $C > 0$ independent of c, with probability at least $1 - \Delta\delta$, we have $\langle\theta^\star, a_s\rangle \geq 1 - 4\varepsilon$ for all $s \in [t, t + \Delta]$, and $m_{t+\Delta} \geq 1 - \varepsilon/2$.*

The main distinction in Lemma 6 is the choice of a smaller $\sigma_s^2$ to encourage exploitation for a small regret: using the local linearity assumption in Assumption 1, the total regret in the epoch is

$$\sum_{s=t}^{t+\Delta}(f(1) - f(\langle\theta^\star, a_s\rangle)) \leq (\Delta + 1) \cdot 4\gamma_2\varepsilon = \widetilde{O}\left(\frac{d^2}{\varepsilon}\right) \quad \text{with probability } 1 - \Delta\delta.$$

In addition, the duration of each epoch becomes longer, with a correspondence $\varepsilon = \widetilde{\Theta}(\frac{d}{\sqrt{t}} \wedge 1)$. This correspondence gives the learning rate and exploration schedule in Theorem 1, as well as the $\widetilde{O}(d\sqrt{T})$ regret bound. The proof of Lemma 6 is deferred to the appendix.

## 4 ANALYSIS OF THE BURN-IN PHASE

The analysis of the SGD dynamics in the burn-in phase relies on similar induction ideas, with a more complicated tradeoff among the three components in the correlation improvement $m_{t+1} - m_t$.

### 4.1 LINK FUNCTION WITH DERIVATIVE LOWER BOUND

We first investigate the simpler scenario in Assumption 2, i.e., $f'(x) \geq c_0$ for all $x \in [0, 1]$. In this case, Theorem 2 is a direct consequence of the following lemma:

**Lemma 7** (Burn-in phase under Assumption 2). *Suppose $m_1 \geq \frac{1}{\sqrt{d}}$. Let $\iota := \log^2(d/\delta)$, and set*

$$\eta_t \equiv \eta := \frac{c}{d\iota}, \quad \sigma_t^2 \equiv \sigma^2 := \gamma_0,$$

*where $c > 0$ is a universal constant. Then for $T := Cd/\eta$ and a large absolute constant $C > 0$ independent of c, we have $m_T \geq 1 - \gamma_0/4$ with probability at least $1 - T\delta$.*

In the sequel we present the proof of Lemma 7. Again we consider the stopping time $T_0 = \min\{t \geq 1 : m_t \geq 1 - \gamma_0/8\}$ and splits into two regimes.

**Regime I:** $t \leq T_0$. If $T_0 > T$, we prove by induction that $m_t \geq \frac{1}{2\sqrt{d}} + c_1\frac{\eta(t-1)}{d}$ for all $t \in [1, T]$ with probability at least $1 - T\delta$, for some absolute constant $c' > 0$ independent of $c$. The base case $t = 1$ is our assumption. Now suppose this lower bound holds for $m_1, \ldots, m_{t-1}$, then by Lemma 1 and 4, with probability at least $1 - \frac{\delta}{4}$, for each $s = 1, \ldots, t - 1$,

$$\left(\mathbb{E}[m_{s+1/2}|\mathcal{F}_s] - m_s\right) + \left(m_{s+1} - m_{s+1/2}\right) = \Omega\left(\frac{\eta}{d}\right) - O\left(\eta^2\log(\frac{2}{\delta})\right) = \Omega\left(\frac{\eta}{d}\right)$$

by our choice of $\eta$. Here we have critically used the condition $m_s = 1 - \Omega(1)$ for $s < T_0$ when applying Lemma 1, and the inductive hypothesis to ensure $m_s > 0$. By Lemma 2 and 3, with probability $1 - \frac{\delta}{4}$, the sum of martingale difference is at most $O(\eta\sqrt{\frac{T}{d}}\log(\frac{T}{\delta})) = O(\sqrt{\eta}\log(\frac{T}{\delta})) \leq \frac{1}{2\sqrt{d}}$ for $c > 0$ small enough. Therefore,

$$m_t \geq m_1 - \frac{1}{2\sqrt{d}} + \sum_{s=1}^{t-1}\Omega\left(\frac{\eta}{d}\right) \geq \frac{1}{2\sqrt{d}} + \Omega\left(\frac{\eta(t-1)}{d}\right),$$

completing the induction step. Now choosing $t = T$ with $C > 0$ large enough shows the opposite result $m_T \geq 1 - \gamma_0/8$, implying that the event $T_0 > T$ only occurs with probability at most $T\delta/2$.

**Regime II:** $T_0 \leq t \leq T$. Under the high-probability event $T_0 \leq T$ and starting from $t = T_0$,

$$m_T - m_{T_0} = \sum_{t=T_0}^{T-1}\left[\underbrace{\left(\mathbb{E}[m_{t+1/2}|\mathcal{F}_t] - m_t\right)}_{\geq 0 \text{ by Lemma 1}} + \underbrace{\left(m_{t+1/2} - \mathbb{E}[m_{t+1/2}|\mathcal{F}_t]\right)}_{=:A_t} + \underbrace{\left(m_{t+1} - m_{t+1/2}\right)}_{=:B_t}\right].$$

By Lemma 2 and 3, $|\sum_{t=T_0}^{T-1} A_t| = O(\eta\sqrt{\frac{T}{d}}\log(\frac{T}{\delta})) = O(\sqrt{\eta}\log(\frac{T}{\delta})) < \frac{\gamma_0}{16}$ with probability $1 - T\delta/2$, for $c > 0$ small enough. In addition, Lemma 4 gives $\sum_{t=T_0}^{T-1}|B_t| = O(T\eta^2\log(\frac{T}{\delta})) = O(d\eta\log(\frac{T}{\delta})) < \frac{\gamma_0}{16}$ with probability $1 - T\delta/2$, again for $c > 0$ small enough. Therefore, at the end of this regime, $m_T \geq m_{T_0} - \gamma_0/8 \geq 1 - \gamma_0/4$ with probability $1 - T\delta$, as desired.

### 4.2 Convex link function

When $f$ is convex in Assumption 3, we establish the following lemma.

**Lemma 8** (Local improvement for convex link function). *For $1 \leq k \leq d - 1$, let $\underline{m}_k := (1 - \gamma_0)^2 \sqrt{k/d}$, and $\overline{m}_k := (1 - \gamma_0/4)\sqrt{k/d}$. Suppose that $m_t \geq \overline{m}_k$ at the beginning of the $k$-th epoch. Let $\iota := \log^2(d/\delta)$, and for $s \geq t$, set*

$$\eta_s \equiv \eta := \frac{cf'(\underline{m}_k)}{\iota d \underline{m}_k}, \quad \sigma_s^2 \equiv \sigma^2 := \gamma_0,$$

*where $c > 0$ is a small absolute constant. Then for $\Delta := Cd(\underline{m}_{k+1} - \underline{m}_k)/(\eta f'(\underline{m}_k))$ and a large absolute constant $C > 0$ independent of $c$, we have $m_{t+\Delta} \geq \overline{m}_{k+1}$ with probability at least $1 - \Delta\delta$.*

Since $m_1 \geq \sqrt{1/d} \geq \overline{m}_1$, a recursive application of Lemma 8 for $k = 1, \ldots, d - 1$ leads to $m_T \geq 1 - \gamma_0/4$ with probability at least $1 - T\delta$, with (recall that $\gamma_0 \leq 0.1$)

$$T = O\left(\log^2\left(\frac{d}{\delta}\right) \cdot d^2 \sum_{k=1}^{d-1} \frac{m_k(m_{k+1} - m_k)}{f'(\underline{m}_k)^2}\right) = \widetilde{O}\left(d^2 \int_{\frac{1}{2\sqrt{d}}}^{1-\gamma_0/4} \frac{x \mathrm{d}x}{f'(x)^2}\right).$$

This completes the proof of Theorem 2. The proof of Lemma 8 is more involved, and we defer the details to the appendix.

## 5 Discussion

**Comparison with other descent algorithms.** Our SGD update in Equation (3) is an online gradient descent applied to the loss $\ell_t(\theta) := \frac{1}{2}(r_t - f(\langle\theta, a_t\rangle))^2$, with $a_t$ chosen according to Equation (2). A typical guarantee in online learning takes the form (e.g., via the sequential Rademacher complexity (Rakhlin et al., 2015))

$$\sum_{t=1}^{T} (f(\langle\theta_t, a_t\rangle) - f(\langle\theta^\star, a_t\rangle))^2 = \widetilde{O}(d).$$

which is known as an online regression oracle (Foster & Rakhlin, 2020; Foster et al., 2021). However, this oracle guarantee alone does not yield the optimal regret of $\theta_t$ in single-index models; see Theorem 1.5 of (Rajaraman et al., 2024) for a general negative result. This motivates us to move beyond standard online learning guarantees and directly analyze the SGD dynamics.

A different descent algorithm for single-index models is also in (Huang et al., 2021), using zeroth-order stochastic optimization to approximate the gradient and implement a noisy power method. In contrast, our SGD is *not* a zeroth-order method: rather than performing gradient descent on the link function $\theta \mapsto f(\langle\theta^\star, \theta\rangle)$ where only a zeroth-order oracle is available, we apply gradient descent to the *population loss* $\theta \mapsto \frac{1}{2}\mathbb{E}(r - f(\langle\theta, a\rangle))^2$ for which an unbiased gradient estimator exists for every $\theta$. This change of objective makes SGD a natural yet novel solution to nonlinear ridge bandits.

**Necessity of monotonicity.** Throughout this paper we assume that the link function $f$ is monotone, an assumption that is not needed in the non-interactive setting (see, e.g., (Ben Arous et al., 2021)). This condition, however, turns out to be essentially necessary for SGD to succeed under our exploration strategy equation 2. Indeed, when $\sigma_t \equiv \sigma$, SGD is performed on the population loss

$$\mathbb{E}\left[(r_t - f(\langle\theta_t, a_t\rangle))^2\right] = \mathbb{E}\left[(f(\langle\theta^\star, a_t\rangle) - f(\langle\theta_t, a_t\rangle))^2\right] + \mathrm{Var}(r_t)$$

$$= \mathbb{E}\left[\left(f\left(\sqrt{1-\sigma^2}\right) - f\left(\sqrt{1-\sigma^2}\langle\theta^\star, \theta_t\rangle + \sigma\langle\theta^\star, Z_t\rangle\right)\right)^2\right] + \mathrm{Var}(r_t)$$

$$\approx \left(f\left(\sqrt{1-\sigma^2}\right) - f\left(\sqrt{1-\sigma^2}\langle\theta^\star, \theta_t\rangle\right)\right)^2 + \mathrm{Var}(r_t),$$

where the last approximation uses that $\langle\theta^\star, Z_t\rangle$ is typically of order $\widetilde{O}(1/\sqrt{d})$ and thus often negligible. Recall that for SGD to succeed at the population level, the population loss must decrease with the alignment $\langle\theta^\star, \theta_t\rangle$ (stated as Assumption A in (Ben Arous et al., 2021)). Treating $\mathrm{Var}(r_t)$

as a constant, this requires $f$ to be increasing on $[0, \sqrt{1 - \sigma^2}]$ in the interactive setting (assuming $f'(0) > 0$). Hence, whenever $\sigma$ is bounded away from 1, a monotonicity assumption on $f$ is indispensable in the interactive setting. By contrast, when $\sigma = 1$ the monotonicity condition is unnecessary: in this case Equation (2) reduces to pure exploration, and the problem essentially collapses to the non-interactive setting. However, this would eliminate the statistical benefits of interaction.

We also provide an explicit counterexample to formally support the above intuition.

**Proposition 1.** *Consider the SGD dynamics in Equation* (3) *applied to the link function*

$$
f(m) = \begin{cases} 0 & \text{if } m \leq 0 \\ -m & \text{if } 0 < m \leq \frac{1}{3} \\ m - \frac{2}{3} & \text{if } \frac{1}{3} < m \leq 1 \end{cases},
$$

*with any initialization $m_1 = \langle \theta^\star, \theta_1 \rangle \leq 0.1$, any exploration schedule $\sigma_t \leq 0.1$, and any learning rate $\eta_t \leq \frac{c}{\log(T/\delta)}$ for some small absolute constant $c > 0$. Then $\mathbb{P}(\max_{t \in [T]} m_t \leq 0.2) \geq 1 - \delta$.*

Note that the above link function $f$ violates the monotonicity condition: it first decreases and then increases on $[0, 1]$. By choosing $\delta = T^{-2}$, Proposition 1 shows that with any practical initialization, any exploration schedule that does not essentially correspond to a non-interactive exploration, and any learning rate that is not too large to escape the local optima, with high probability the resulting SGD cannot achieve an alignment better than a small constant (say 0.2).

**Comparison with information exponent.** In the non-interactive case with $a_t \sim \mathcal{N}(0, I_d)$, it is known that the *information exponent* of $f$ determines the sample complexity of SGD. In the interactive case, however, the monotonicity of $f$ ensures that the information exponent is always 1. Indeed, for the first Hermite polynomial $H_1(x) = x$, Chebyshev's sum inequality yields

$$
\mathbb{E}_{Z \sim \mathcal{N}(0,1)}[f(Z) H_1(Z)] \geq \mathbb{E}_{Z \sim \mathcal{N}(0,1)}[f(Z)] \cdot \mathbb{E}_{Z \sim \mathcal{N}(0,1)}[H_1(Z)] = 0,
$$

with equality iff $f \equiv c$ is a constant. Moreover, the sample complexity predicted by the information exponent is no longer tight. For instance, when $f(x) = x^p$ with an odd $p \geq 3$, the sample complexity of SGD with $a_t \sim \mathcal{N}(0, I_d/d)$ is $\widetilde{O}(d^{p+1})$ (see remark below), which is strictly worse than the $\widetilde{O}(d^p)$ guarantee obtained by our interactive SGD. These observations show that the information exponent ceases to be an informative measure for SGD in the interactive case, for the actions $a_t$ are no longer Gaussian.

**Remark 2.** *For $f(x) = x^p$ with odd $p \geq 3$, the population square loss has information exponent equal to 1. Let $c_1$ be the coefficient of the linear term $\langle \theta^\star, \theta_t \rangle$ in*

$$
\mathbb{E}_{X \sim \mathcal{N}(0, I_d)} \Big[ (f(\langle \theta^\star, X \rangle) - f(\langle \theta_t, X \rangle))^2 \Big],
$$

*then $c_1 = -2u_1(f)^2$ with $u_1(f)$ being the first Hermite coefficient of $f$. When we scale down the input features into $X \sim \mathcal{N}(0, I_d/d)$, we effectively change $f$ to $\widetilde{f}(x) = (x/\sqrt{d})^p$, so $c_1$ becomes $d^{-p}c_1$. Therefore, the SNR effectively worsens by a factor of $d^p$.*

**Dropping the convexity assumption.** The convexity assumption in Assumption 3 is not required in the statistical complexity framework developed for ridge bandits in (Rajaraman et al., 2024). Relying only on the monotonicity of $f$, they establish the upper bound

$$
\widetilde{O}\Big( d^2 \int_{1/\sqrt{d}}^{1/2} \frac{\mathrm{d}[x^2]}{\max_{\frac{1}{\sqrt{d}} \leq y \leq x} f'(y)^2} \Big)
$$

on the sample complexity of finding an action $a_t$ with $\langle \theta^\star, a_t \rangle \geq 1/2$. In comparison, under our convexity assumption the denominator simplifies to $f'(x)^2$. There are two main obstacles to recovering this sharper bound. First, our analysis in Lemma 4 requires a conservative choice of the learning rate $\eta_t$, which in turn depends on having a lower bound for $f'(m_t)$ at the current correlation $m_t$. Obtaining such a bound is challenging without further conditions on $f$. In this paper we handle this by using $f'(m_t) \geq c$ in the generalized linear case, and $f'(m_t) \geq f'(\underline{m}_t)$ in the convex case, where $\underline{m}_t \leq m_t$ is known. Second, achieving the factor $\max_{1/\sqrt{d} \leq y \leq x} f'(y)^2$ requires a careful tuning of $\sigma_t$ to target the maximizer of $f'$, which in turn relies on knowledge of the current correlation $m_t$. In (Rajaraman et al., 2024), this is accomplished by running a separate hypothesis test. However, such an additional testing step is not compatible with the dynamics of SGD.

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

## A PICTORIAL ILLUSTRATION OF LEMMA 5

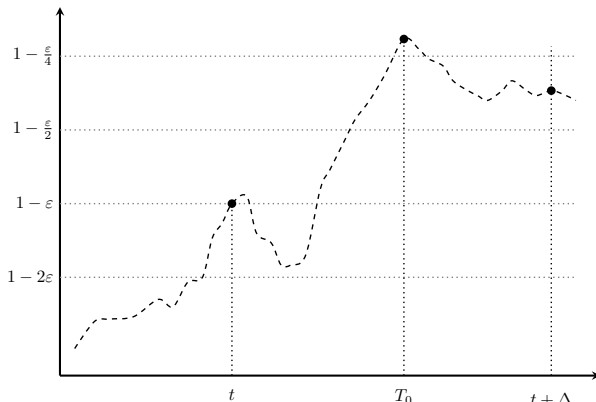

Figure 2: *An example behavior of SGD for pure exploration in the learning phase (cf. Lemma 5).* For appropriately chosen learning rates, if the correlation hits $m_t \geq 1 - \varepsilon$ at time $t$, the SGD dynamics will enjoy the following behaviors with high probability: $(i)$ the trajectory will never degrade too significantly, satisfying $m_s \geq 1 - 2\varepsilon$ for all $t \leq s \leq t + \Delta$; $(ii)$ at some time $s = T_0 \in [t, t + \Delta]$, $m_s$ improves to at least $1 - \frac{\varepsilon}{4}$; and $(iii)$ thereafter, $m_s$ may decrease, but will never fall below $1 - \frac{\varepsilon}{2}$ for all $T_0 \leq s \leq t + \Delta$.

## B PROOFS OF MAIN LEMMAS

### B.1 PROOF OF COROLLARY 1

By Theorem 1 and 2, it remains to show that both the initialization cost $\widetilde{O}((f(1/\sqrt{d}) - f(0))^{-2})$ and the burn-in cost $\widetilde{O}(d^2)$ under Assumption 2 are dominated by the integral.

For the initialization cost, we have

$$\frac{1}{(f(\frac{1}{\sqrt{d}}) - f(0))^2} \overset{(a)}{\leq} \frac{1}{(f(\frac{1}{\sqrt{d}}) - f(\frac{1}{2\sqrt{d}}))^2} = \frac{4d}{\left(2\sqrt{d} \int_{1/(2\sqrt{d})}^{1/\sqrt{d}} f'(m)\mathrm{d}m\right)^2}$$

$$\overset{(b)}{\leq} 4d \cdot 2\sqrt{d} \int_{1/(2\sqrt{d})}^{1/\sqrt{d}} \frac{1}{f'(m)^2}\mathrm{d}m \leq 16d^2 \int_{1/(2\sqrt{d})}^{1/\sqrt{d}} \frac{m}{f'(m)^2}\mathrm{d}m,$$

where (a) follows from the monotonicity of $f$, and (b) applies Jensen's inequality.

For the burn-in cost $\widetilde{O}(d^2)$ under Assumption 2, we simply note that $f'(x) \leq \gamma_2$ when $x \in [1-\gamma_0, 1]$ by Assumption 1, so that

$$d^2 \int_{1-\gamma_0}^{1-\gamma_0/4} \frac{m}{f'(m)^2}\mathrm{d}m \geq d^2 \cdot \frac{3\gamma_0}{4} \frac{1-\gamma_0}{\gamma_2^2} = \Omega(d^2).$$

These complete the proof.

### B.2 PROOF OF LEMMA 1

Observe that

$$\mathbb{E}[\theta_{t+1/2}|\mathcal{F}_t] = \mathbb{E}\left[\theta_t - \eta_t\sigma_t\left[(f(\langle a_t, \theta_t\rangle) - f(\langle a_t, \theta^\star\rangle) - N_t)f'(\langle a_t, \theta_t\rangle)\right] \cdot Z_t|\mathcal{F}_t\right]$$
$$= \theta_t - \eta_t\sigma_t\mathbb{E}\left[\left[(f(\langle a_t, \theta_t\rangle) - f(\langle a_t, \theta^\star\rangle))f'(\langle a_t, \theta_t\rangle)\right] \cdot Z_t|\mathcal{F}_t\right].$$

Recall that $a_t = \sqrt{1 - \sigma_t^2}\,\theta_t + \sigma_t Z_t$ in Equation (2). Since $Z_t \perp \theta_t$ almost surely, $\langle a_t, \theta_t\rangle = \sqrt{1 - \sigma_t^2}$. Taking an inner product with $\theta^\star$ on both sides,

$$\mathbb{E}[m_{t+1/2}|\mathcal{F}_t] - m_t = \eta_t\sigma_t f'\left(\sqrt{1 - \sigma_t^2}\right) \cdot \mathbb{E}\left[f\left(\sqrt{1 - \sigma_t^2}\,\langle\theta_t, \theta^\star\rangle + \sigma_t\langle Z_t, \theta^\star\rangle\right)\langle Z_t, \theta^\star\rangle\Big|\mathcal{F}_t\right].$$

Since $Z_t \sim \mathrm{Unif}(\{x \in \mathbb{S}^{d-1} : x \perp \theta_t\})$, the random variable $(1-m_t^2)^{-1/2}\langle Z_t, \theta^\star \rangle$ is distributed as the one-dimensional marginal of a uniform random vector on $\mathbb{S}^{d-2}$; denote by $X$ a random variable following this distribution. Consequently, for

$$g(x) = f\left( \sqrt{1-\sigma_t^2} \, \langle \theta_t, \theta^\star \rangle + \sigma_t \sqrt{1-m_t^2} x \right),$$

an application of the spherical Stein's lemma (cf. Lemma 10) gives

$$\begin{aligned}
&\mathbb{E}[m_{t+1/2}|\mathcal{F}_t] - m_t \\
&= \eta_t \sigma_t f'\left(\sqrt{1-\sigma_t^2}\right)\sqrt{1-m_t^2} \cdot \mathbb{E}\left[g\left(X\right)X|\mathcal{F}_t\right] \\
&= \frac{\eta_t \sigma_t}{d-2} f'\left(\sqrt{1-\sigma_t^2}\right)\sqrt{1-m_t^2} \cdot \mathbb{E}\left[g'\left(X\right)\left(1-X^2\right)\middle|\mathcal{F}_t\right] \\
&= \frac{\eta_t \sigma_t^2}{d-2} f'\left(\sqrt{1-\sigma_t^2}\right)(1-m_t^2) \cdot \mathbb{E}\left[f'\left(\sqrt{1-\sigma_t^2}\,m_t + \sigma_t \sqrt{1-m_t^2}X\right)(1-X^2)\middle|\mathcal{F}_t\right].
\end{aligned}$$

This is the desired identity. For the other inequalities, under Assumption 2 and $m_t \geq 0$, for

$$h(x) = f'\left( \sqrt{1-\sigma_t^2} \, \langle \theta_t, \theta^\star \rangle + \sigma_t \sqrt{1-m_t^2}x \right) \geq 0,$$

we have

$$\begin{aligned}
\mathbb{E}[h(X)(1-X^2)] &\geq \mathbb{E}[h(X)(1-X^2)\mathbb{1}(X \geq 0)] \\
&\geq c_0 \mathbb{E}[(1-X^2)\mathbb{1}(X \geq 0)] = c_0 \cdot \frac{d-2}{2(d-1)} = \Omega(1)
\end{aligned}$$

for $d \geq 3$. Under Assumption 3 and $m_t \geq 0$, we then write

$$\begin{aligned}
\mathbb{E}[h(X)(1-X^2)] &\geq \mathbb{E}[h(X)(1-X^2)\mathbb{1}(X \geq 0)] \\
&\geq f'(\sqrt{1-\sigma_t^2}\,m_t) \cdot \mathbb{E}[(1-X^2)\mathbb{1}(X \geq 0)] \\
&= \Omega(f'(\sqrt{1-\sigma_t^2}\,m_t)).
\end{aligned}$$

### B.3 PROOF OF LEMMA 2

By definition,

$$m_{t+1/2} - m_t = \eta_t \sigma_t (f(\langle a_t, \theta^\star \rangle) + N_t - f(\langle a_t, \theta_t \rangle))f'(\langle a_t, \theta_t \rangle) \cdot \langle Z_t, \theta^\star \rangle$$

Define two new random variables:

$$\begin{aligned}
\xi^{(1)} &= \eta_t \sigma_t (f(\langle a_t, \theta^\star \rangle) - f(\langle a_t, \theta_t \rangle))f'(\langle a_t, \theta_t \rangle) \cdot \langle Z_t, \theta^\star \rangle, \\
\xi^{(2)} &= \eta_t \sigma_t N_t f'(\langle a_t, \theta_t \rangle) \cdot \langle Z_t, \theta^\star \rangle,
\end{aligned}$$

such that $m_{t+1/2} - m_t = \xi^{(1)} + \xi^{(2)}$. We will show that each of these random variables is subexponential with a bounded $\Psi_1$-Orlicz norm.

For $\xi^{(1)}$, note that $|f(\langle a_t, \theta^\star \rangle) - f(\langle a_t, \theta_t \rangle)| \leq 2\|f\|_\infty$ and $\langle a_t, \theta_t \rangle = \sqrt{1-\sigma_t^2}$. In addition,

$$\langle Z_t, \theta^\star \rangle \stackrel{d}{=} \sqrt{1-m_t^2}X,$$

where $X$ follows the one-dimensional marginal of a uniform random vector on $\mathbb{S}^{d-2}$. By Lemma 11, it holds that $\|X\|_{\Psi_2} \leq \|\mathcal{N}(0, d^{-1})\|_{\Psi_2} = O(d^{-1/2})$. Therefore,

$$\|\xi^{(1)}\|_{\Psi_1} \stackrel{(a)}{=} O(\|\xi^{(1)}\|_{\Psi_2}) = O\left( \frac{\eta_t \sigma_t f'(\sqrt{1-\sigma_t^2})\sqrt{1-m_t^2}}{\sqrt{d}} \right),$$

where (a) follows from (Vershynin, 2018, Remark 2.8.8).

For $\xi^{(2)}$, note that $\|N_t\|_{\Psi_2} \leq 1$ by the 1-subGaussian assumption on the noise. Therefore, by independence of $Z_t$ and $N_t$, (Vershynin, 2018, Lemma 2.8.6) gives

$$\|\xi^{(2)}\|_{\Psi_1} \leq \eta_t \sigma_t f'(\sqrt{1-\sigma_t^2})\|N_t\|_{\Psi_2}\|\langle Z_t, \theta^\star\rangle\|_{\Psi_2} = O\left(\frac{\eta_t \sigma_t f'(\sqrt{1-\sigma_t^2})\sqrt{1-m_t^2}}{\sqrt{d}}\right).$$

Finally, the triangle inequality of the $\Psi_1$ norm gives

$$\|m_{t+1/2} - \mathbb{E}[m_{t+1/2}|\mathcal{F}_t]\|_{\Psi_1} \leq \|\xi^{(1)}\|_{\Psi_1} + \|\xi^{(2)}\|_{\Psi_1} = O\left(\frac{\eta_t \sigma_t f'(\sqrt{1-\sigma_t^2})\sqrt{1-m_t^2}}{\sqrt{d}}\right).$$

### B.4 Proof of Lemma 3

For notational simplicity we write $S_t := S_t^{t_0, \beta}$. By Lemma 2,

$$\log \mathbb{E}[\exp(\lambda(S_{t+1} - S_t))|\mathcal{F}_t] \leq C\beta^{2(t-t_0)}K_t^2 \lambda^2, \quad \text{for all } |\lambda| \leq \frac{1}{C\beta^{t-t_0}K_t}.$$

Here $C > 0$ is a universal constant. We show that $K_t \leq 1$ almost surely. In fact, $f'(\sqrt{1-\sigma_t^2}) \leq \gamma_2$ by Assumption 1 when $\sigma_t^2 \leq \gamma_0$, and

$$K_t \leq C_{\mathsf{se}}\eta_t \gamma_2 \leq 1$$

by the choice of $\eta_t$. Consequently, for $V_t = C\sum_{s=t_0}^{t-1}\beta^{2(s-t_0)}K_s^2$, $\lambda_{\max} := \frac{1}{2C}$, and

$$\psi(\lambda) = \frac{\lambda^2}{1 - \lambda/\lambda_{\max}}, \quad \lambda \in [0, \lambda_{\max}),$$

it holds that

$$\mathbb{E}[\exp(\lambda S_{t+1} - \psi(\lambda)V_{t+1})|\mathcal{F}_t] \leq \exp(\lambda S_t - \psi(\lambda)V_t), \quad \lambda \in [0, \lambda_{\max}), \beta^{t-t_0} \leq 2.$$

Therefore, the conditions of Lemma 9 are fulfilled, and the claimed upper tail of $S_t$ follows from choosing $\omega = 1$. Replacing $S_t$ by $-S_t$ in the above analysis gives the lower tail of $S_t$.

### B.5 Proof of Lemma 4

Since $\theta_t \perp Z_t$, the iterate $\theta_{t+1/2}$ in Equation (3) satisfies

$$\|\theta_{t+1/2}\|^2 = 1 + \eta_t^2 \sigma_t^2 f'(\sqrt{1-\sigma_t^2})^2(f(\langle\theta_t, a_t\rangle) - r_t)^2.$$

Therefore, $\|\theta_{t+1/2}\| \geq 1$, it is clear that

$$\begin{aligned}
m_{t+1} = \frac{m_{t+1/2}}{\|\theta_{t+1/2}\|} &= m_{t+1/2} - \frac{m_{t+1/2}}{\|\theta_{t+1/2}\|}\left(\|\theta_{t+1/2}\| - 1\right) \\
&\geq m_{t+1/2} - \frac{1}{2}\eta_t^2 \sigma_t^2 f'(\sqrt{1-\sigma_t^2})^2(f(\langle\theta_t, a_t\rangle) - r_t)^2,
\end{aligned}$$

using $\sqrt{1+x} - 1 \leq \frac{x}{2}$ for $x \geq 0$, and $|m_{t+1/2}|/\|\theta_{t+1/2}\| \leq 1$. The first statement now follows from the sub-Gaussian concentration of $r_t$, which implies $(f(\langle\theta_t, a_t\rangle) - r_t)^2 = O(\log(1/\delta))$ with probability at least $1 - \delta$.

For the second statement, $m_{t+1} \leq m_{t+1/2}$ follows from $\|\theta_{t+1/2}\| \geq 1$. The other direction follows from the same high-probability upper bound of $\|\theta_{t+1/2}\| - 1$, and the simple inequality $\frac{1}{1+x} \geq 1 - x$ for $x \geq 0$.

### B.6 Proof of Lemma 6

As we showed in the proof of Lemma 5, we will show by induction on $s$ that with probability at least $1 - \Delta\delta/3$, $m_s \geq m_t - \frac{\varepsilon}{4}$ for all $s \in [t, t+\Delta]$. The base case $s = t$ is ensured by the assumption

$m_t \geq 1 - \varepsilon$. For the inductive step, the induction hypothesis implies that $m_t, \ldots, m_{s-1} \geq 1 - 2\varepsilon$. Then

$$m_s - m_t = \sum_{r=t}^{s-1} \Big[ \underbrace{\big(\mathbb{E}[m_{r+1/2}|\mathcal{F}_r] - m_r\big)}_{\geq 0 \text{ by Lemma 1}} + \underbrace{\big(m_{r+1/2} - \mathbb{E}[m_{r+1/2}|\mathcal{F}_r]\big)}_{=:A_r} + \underbrace{\big(m_{r+1} - m_{r+1/2}\big)}_{=:B_r} \Big].$$

Thanks to the inductive hypothesis, $K_r = O(\eta \frac{\varepsilon}{\sqrt{d}})$ for all $r \in [t, s-1]$ in Lemma 2, so Lemma 3 (with $t_0 = t, \beta = 1$) gives

$$\big| \sum_{r=t}^{s-1} A_r \big| = O(\eta \sqrt{\frac{\Delta \varepsilon^2}{d}} \log(\frac{\Delta}{\delta})) = O(\sqrt{\eta \varepsilon} \log(\frac{\Delta}{\delta})) < \frac{\varepsilon}{8}$$

with probability $1 - \frac{\delta}{6}$, by choosing $c > 0$ small enough. Similarly,

$$\sum_{r=t}^{s-1} |B_r| = O(\Delta \eta^2 \varepsilon \log(\frac{\Delta}{\delta})) = O(d\eta \log(\frac{\Delta}{\delta})) < \frac{\varepsilon}{8}$$

with probability $1 - \frac{\delta}{6}$, by Lemma 4 and choosing $c > 0$ small enough. This implies that $m_s \geq m_t - \frac{\varepsilon}{4}$ with probability $1 - \frac{\delta}{3}$, completing the induction.

Conditioned on the event $m_s \geq 1 - 2\varepsilon$ for all $s \in [t, t+\Delta]$, we distinguish into two regimes in this epoch. Let $T_0 \geq t$ be the stopping time where $m_s > 1 - \varepsilon/4$ for the first time.

**Regime I:** $t \leq s < T_0$. In this regime $m_s \in [1 - 2\varepsilon, 1 - \varepsilon/4]$. We show that $T_0 \leq t + \Delta$ with probability $1 - \Delta\delta/3$. If $T_0 > t + \Delta$, using the same high-probability bounds, we have

$$m_{t+\Delta} - m_t \geq \sum_{s=t}^{t+\Delta-1} \big(\mathbb{E}[m_{s+1/2}|\mathcal{F}_s] - m_s\big) - \frac{\varepsilon}{4}$$

with probability $1 - \Delta\delta/3$. By Lemma 1 with $1 - m_s^2 = \Omega(\varepsilon)$ and $\sqrt{1 - \sigma_s^2} m_s \geq 1 - \gamma_0$ for $s < T_0$, the total drift is $\Omega(\frac{\Delta\eta\varepsilon^2}{d}) = \Omega(C\varepsilon)$. Therefore, for a large absolute constant $C > 0$, we would have $m_{t+\Delta} \geq 1 - \varepsilon/2$, a contradiction to the assumption $T_0 > t + \Delta$.

**Regime II:** $s \geq T_0$. As shown above, this regime is non-empty with high probability. The same induction starting from $s = T_0$ shows that, with probability $1 - \Delta\delta/3$, $m_s \geq m_{T_0} - \varepsilon/4$ holds for all $s \in [T_0, t + \Delta]$. In particular, choosing $s = t + \Delta$ gives the desired result $m_{t+\Delta} \geq 1 - \varepsilon/2$.

Finally, to lower bound $\langle \theta^\star, a_s \rangle$ during this epoch, we simply note that

$$\begin{aligned}
\langle \theta^\star, a_s \rangle &= \sqrt{1 - \sigma_s^2} m_s + \sigma_s \langle \theta^\star, Z_s \rangle \\
&= \sqrt{1 - \sigma_s^2} m_s + \sigma_s \langle \theta^\star - m_s \theta_s, Z_s \rangle \\
&\geq \sqrt{1 - \sigma_s^2} m_s - \sigma_s \| \theta^\star - m_s \theta_s \| \\
&= \sqrt{1 - \sigma_s^2} m_s - \sigma_s \sqrt{1 - m_s^2}.
\end{aligned}$$

Under the good event $m_s \geq m_t - \frac{\varepsilon}{4} \geq 1 - \frac{3\varepsilon}{2}$, by $\sigma_s \equiv \sqrt{\varepsilon}$ we have $\langle \theta^\star, a_s \rangle \geq 1 - 4\varepsilon$, as desired.

### B.7 PROOF OF LEMMA 8

Let

$$\beta := 1 - C_{\mathsf{nm}} \gamma_2^2 \eta^2 \sigma^2 \log\left(\frac{4\Delta}{\delta}\right), \tag{4}$$

with $C_{\mathsf{nm}}$ given in Lemma 4. By the choice of $\eta$, when the constant $c > 0$ is small enough, we have $\beta \in (1/2, 1)$. In addition, let

$$T_0 = \min\left\{ s \geq t : m_s \geq \left(1 - \frac{\gamma_0}{8}\right) \sqrt{\frac{k+1}{d}} \right\} \tag{5}$$

be the stopping time when the correlation $m_s$ first hits a given threshold. Unlike the other proofs, the event $T_0 \leq t + \Delta$ no longer occurs with high probability, and our proof will discuss both cases.

**Case I:** $T_0 > t + \Delta$. Define the following event:

$$\mathcal{E}_s := \left\{ m_s \geq \overline{m}_k - \frac{\gamma_0}{d} + \frac{c'\eta f'(\underline{m}_k)}{d}(s-t) \right\}, \tag{6}$$

where $c' > 0$ is a small absolute constant (to be chosen later) independent of $c$. We will prove by induction that

$$\mathbb{P}\left( (\cup_{r=t}^s \mathcal{E}_r^c) \cap \{T_0 > t + \Delta\} \right) \leq (s-t)\frac{\delta}{2}, \quad \text{for all } s = t, t+1, \ldots, t+\Delta. \tag{7}$$

The base case follows from the assumption $m_t \geq \overline{m}_k$, so that $\mathbb{P}(\mathcal{E}_t^c) = 0$. For the inductive step, suppose that Equation (7) holds for $s - 1$. Since $\mathbb{P}(A \cup B) = \mathbb{P}(A) + \mathbb{P}(A^c \cap B)$, it suffices to prove that

$$\mathbb{P}\left( \mathcal{E}_s^c \cap \left( \cap_{r=t}^{s-1} \mathcal{E}_r \right) \cap \{T_0 > t + \Delta\} \right) \leq \frac{\delta}{2}. \tag{8}$$

To this end, we introduce some additional events. First, applying Lemma 3 with $t_0 = t$ and $\beta^{-1} \leq 2$ in Equation (4) gives

$$\mathbb{P}(\mathcal{E}_{s,1}) := \mathbb{P}\left( \left| \sum_{r=t}^s \frac{m_{r+1/2} - \mathbb{E}[m_{r+1/2}|\mathcal{F}_r]}{\beta^{r-t}} \right| \leq C\eta \sqrt{\frac{\Delta}{d}} \log\left(\frac{d}{\delta}\right) \right) \geq 1 - \frac{\delta}{4\Delta}, \tag{9}$$

for some absolute constant $C > 0$. To see Equation (9), note that

$$\beta^{-\Delta} = \exp\left( O((1-\beta)\Delta) \right) = \exp\left( O\left( \eta^2 \Delta \log \frac{d}{\delta} \right) \right) = \exp\left( O\left( \frac{cC}{\iota d} \right) \right) = 1 + \frac{o_c(1)}{d}, \tag{10}$$

so that the condition $\beta^{-\Delta} \leq 2$ holds for small $c > 0$, and $\sum_{r=t}^s \beta^{-2(r-t)} = O(s-t+1) = O(\Delta)$. In addition, let $\mathcal{E}_{s,2}$ be the good event that the lower bound in Lemma 4 holds for $m_{s+1}$, with $\delta/(4\Delta)$ in place of $\delta$.

Note that $\mathcal{E}_r \cap \mathcal{E}_{r,2} \cap \{T_0 > t + \Delta\}$ implies that

$$m_{r+1} \geq \beta m_{r+1/2}$$
$$= \beta \left( m_{r+1/2} - \mathbb{E}[m_{r+1/2}|\mathcal{F}_t] + \mathbb{E}[m_{r+1/2}|\mathcal{F}_t] - m_r + m_r \right)$$
$$\geq \beta \left( m_{r+1/2} - \mathbb{E}[m_{r+1/2}|\mathcal{F}_t] + c_1 \frac{\eta f'(\underline{m}_k)}{d} + m_r \right),$$

where $c_1 > 0$ is an absolute constant, and the last step invokes Lemma 1, uses $m_r \leq 1 - \Omega(1)$ since $r \leq t + \Delta < T_0$, and

$$\sqrt{1 - \sigma^2} m_r \geq \sqrt{1 - \gamma_0}\left( \overline{m}_k - \frac{\gamma_0}{d} \right) \geq (1 - \gamma_0)^2 \sqrt{\frac{k}{d}} = \underline{m}_k$$

by Equation (6) and the definitions of $\overline{m}_k, \underline{m}_k$. Summing over $r = t, \ldots, s-1$, the event $\cap_{r=t}^{s-1}(\mathcal{E}_r \cap \mathcal{E}_{r,2}) \cap \{T_0 > t + \Delta\}$ implies that

$$m_s \geq \beta^{s-t}\left( m_t + \sum_{r=t}^{s-1} \frac{m_{r+1/2} - \mathbb{E}[m_{r+1/2}|\mathcal{F}_t]}{\beta^{r-t}} \right) + c_1 \frac{\eta f'(\underline{m}_k)}{d} \sum_{r=t}^{s-1} \beta^{r+1-t}.$$

In view of Equation (9) and Equation (10), a further intersection with $\mathcal{E}_{s-1}$ implies that

$$m_s \geq \left( 1 - \frac{o_c(1)}{d} \right)\overline{m}_k - C\eta\sqrt{\frac{\Delta}{d}} \log\left(\frac{d}{\delta}\right) + \frac{c'\eta f'(\underline{m}_k)}{d}(s-t)$$
$$= \left( 1 - \frac{o_c(1)}{d} \right)\overline{m}_k - O\left(\frac{cC}{d}\right) + \frac{c'\eta f'(\underline{m}_k)}{d}(s-t)$$
$$\geq \overline{m}_k - \frac{\gamma_0}{d} + \frac{c'\eta f'(\underline{m}_k)}{d}(s-t)$$

for $c > 0$ small enough; this is precisely the event $\mathcal{E}_s$. In other words, we have shown that

$$\mathcal{E}_s^c \cap \left( \cap_{r=t}^{s-1}(\mathcal{E}_r \cap \mathcal{E}_{r,1} \cap \mathcal{E}_{r,2}) \right) \cap \{T_0 > t + \Delta\} = \varnothing. \tag{11}$$

By Equation (11), we have

$$\mathbb{P}\left(\mathcal{E}_s^c \cap \left(\cap_{r=t}^{s-1}\mathcal{E}_r\right) \cap \{T_0 > t+\Delta\}\right)$$
$$\leq \mathbb{P}(\cup_{r=t}^{s-1}\mathcal{E}_{r,1}^c) + \mathbb{P}\left(\left(\cup_{r=t}^{s-1}\mathcal{E}_{r,2}^c\right) \cap \left(\cap_{r=t}^{s-1}(\mathcal{E}_r \cap \mathcal{E}_{r,1})\right) \cap \{T_0 > t+\Delta\}\right).$$

By Equation (9) and the union bound, the first probability is at most $\frac{\delta}{4}$. For the second probability, the same program above shows that $(\cap_{i=t}^{r-1}\mathcal{E}_{i,2}) \cap (\cap_{i=t}^{r}(\mathcal{E}_r \cap \mathcal{E}_{r,1})) \cap \{T_0 > t+\Delta\}$ implies $m_{r+1/2} \geq 0$, which is the prerequisite of Lemma 4. Therefore, the conditional probability of $\mathcal{E}_{r,2}$ is at least $1 - \frac{\delta}{4\Delta}$, and by a union bound the second probability is at most $\frac{\delta}{4}$. This proves Equation (8) and completes the induction.

Finally, note that $\mathcal{E}_{t+\Delta}$ implies that

$$m_{t+\Delta} \geq \overline{m}_k - \frac{\gamma_0}{d} + \frac{c'\eta f'(\underline{m}_k)}{d}\Delta$$
$$= \overline{m}_k - \frac{\gamma_0}{d} + c'C(\underline{m}_{k+1} - \underline{m}_k) \geq \overline{m}_{k+1},$$

by choosing $C > 0$ large enough. Therefore, Equation (7) with $s = t + \Delta$ implies that

$$\mathbb{P}(\{m_{t+\Delta} < \overline{m}_{k+1}\} \cap \{T_0 > t+\Delta\}) \leq \frac{\Delta\delta}{2}. \tag{12}$$

**Case II: $T_0 \leq t + \Delta$.** We apply our usual program to this case: if $T_0 \leq t + \Delta$, then

$$m_{t+\Delta} - m_{T_0} = \sum_{s=T_0}^{t+\Delta-1} \left[ \underbrace{\left(\mathbb{E}[m_{s+1/2}|\mathcal{F}_s] - m_t\right)}_{\geq 0 \text{ by Lemma 1}} + \underbrace{\left(m_{s+1/2} - \mathbb{E}[m_{s+1/2}|\mathcal{F}_s]\right)}_{=:A_s} + \underbrace{\left(m_{s+1} - m_{s+1/2}\right)}_{=:B_s} \right].$$

By Lemma 3, with probability at least $1 - \frac{\Delta\delta}{4}$,

$$\left|\sum_{s=T_0}^{t+\Delta-1} A_s\right| = O\left(\eta\sqrt{\frac{\Delta}{d}}\log\left(\frac{d}{\delta}\right)\right) = O\left(\frac{cC}{d}\right).$$

By Lemma 4, with probability at least $1 - \frac{\Delta\delta}{4}$,

$$\sum_{s=T_0}^{t+\Delta-1} |B_s| = O\left(\Delta \cdot \eta^2 \log\left(\frac{d}{\delta}\right)\right) = O\left(\frac{cC}{d}\right).$$

Therefore, conditioned on $T_0 \leq t + \Delta$, with probability at least $1 - \frac{\Delta\delta}{2}$,

$$m_{t+\Delta} \geq \left(1 - \frac{\gamma_0}{8}\right)\sqrt{\frac{k}{d}} - O\left(\frac{cC}{d}\right) \geq \left(1 - \frac{\gamma_0}{4}\right)\sqrt{\frac{k}{d}} = \overline{m}_k$$

for a small enough constant $c > 0$. In other words,

$$\mathbb{P}(\{m_{t+\Delta} < \overline{m}_{k+1}\} \cap \{T_0 \leq t+\Delta\}) \leq \frac{\Delta\delta}{2}. \tag{13}$$

Finally, a combination of Equation (12) and Equation (13) gives $\mathbb{P}(m_{t+\Delta} < \overline{m}_{k+1}) \leq \Delta\delta$, which is the desired result.

### B.8 PROOF OF PROPOSITION 1

Let $T_0$ be the first time $t \geq 1$ such that $m_t \geq 0.1$. If $T_0 > T$, the target claim $\max_{t\in[T]} m_t \leq 0.2$ is clearly true. Hence in the sequel we condition on the event $T_0 \leq T$. In addition, by Gaussian tail bounds, we have $\max_{t\in[T]} |r_t| = O(\sqrt{\log(T/\delta)})$ with probability at least $1 - \delta/4$. By Equation (3), we then have a deterministic inequality

$$m_{T_0-1/2} \leq m_{T_0-1} + C\eta_{T_0-1}\sqrt{\log(T/\delta)} \leq m_{T_0-1} + 0.05 \leq 0.15,$$

by assumption of $\eta_t \leq \frac{c}{\log(T/\delta)}$ for a sufficiently small constant $c > 0$, and the definition of $T_0$ that $m_{T_0-1} \leq 0.1$. By Lemma 4, this implies that $m_{T_0} \leq 0.15$.

In the sequel, we start from $m_{T_0} \in [0.1, 0.15]$, and for notational simplicity we redefine $m_{T_0}$ to be our starting point, i.e. $T_0 = 1$. Next we consider the time interval $[1, T_1]$ with

$$T_1 = \min\left\{ t \geq 1 : \sum_{s \leq t} \frac{\eta_s^2 \sigma_s^2}{d} \geq \frac{c_1}{\log^2(T/\delta)} \right\},$$

for some absolute constant $c_1 > 0$ to be chosen later. We prove the following claims.

**Claim I:** $\max_{t \in [T_1]} m_t \leq 0.2$ **with probability at least** $1 - \delta T_1/(4T)$.    To prove this claim, we first show that when $m_t \leq 0.2$, then

$$\mathbb{E}[m_{t+1/2}|\mathcal{F}_t] \leq m_t. \tag{14}$$

Indeed, by Lemma 1,

$$\mathbb{E}[m_{t+1/2}|\mathcal{F}_t] - m_t$$
$$= \frac{\eta_t \sigma_t^2}{d - 2} f'\left(\sqrt{1 - \sigma_t^2}\right)(1 - m_t^2) \cdot \mathbb{E}\left[ f'\left(\sqrt{1 - \sigma_t^2}\, m_t + \sigma_t \sqrt{1 - m_t^2} X\right)(1 - X^2) \Big| \mathcal{F}_t \right].$$

Since $\sigma_t \leq 0.1, m_t \leq 0.2$, and $|X| \leq 1$ almost surely, we have

$$\sqrt{1 - \sigma_t^2}\, m_t + \sigma_t \sqrt{1 - m_t^2} X \leq m_t + \sigma_t \leq 0.3 < \frac{1}{3}.$$

Since $f'(m) \leq 0$ for all $m \leq 1/3$ in our construction, and $f'(\sqrt{1 - \sigma_t^2}) > 0$, we obtain Equation (14).

Next, without loss of generality we assume that $m_t \geq 0$ for all $t \in [T_1]$, since a negative $m_t$ only makes the target claim simpler. For every $t \in [T_1]$,

$$m_t - m_1 = \sum_{r=1}^{t-1}\left[ \underbrace{\left(\mathbb{E}[m_{r+1/2}|\mathcal{F}_r] - m_r\right)}_{\leq 0 \text{ by Equation (14)}} + \underbrace{\left(m_{r+1/2} - \mathbb{E}[m_{r+1/2}|\mathcal{F}_r]\right)}_{=:A_r} + \underbrace{\left(m_{r+1} - m_{r+1/2}\right)}_{\leq 0 \text{ by Lemma 4}} \right].$$

By Lemma 3 with $\beta = 1$, we get

$$\left| \sum_{r=1}^{t-1} A_r \right| \leq C \log\left(\frac{T}{\delta}\right) \sqrt{\sum_{r=1}^{t-1} \frac{\sigma_r^2 \eta_r^2}{d}}$$

with probability at least $1 - \delta/(4T)$, for some absolute constant $C > 0$. By the definition of $T_1$, we obtain $|\sum_{r=1}^{t-1} A_r| \leq 0.05$ for a sufficiently small $c_1 > 0$. Therefore, $m_t \leq m_1 + 0.05 \leq 0.2$ with probability at least $1 - \delta/(4T)$, and an induction on $t$ with a union bound gives the target claim.

**Claim II:** $\min_{t \in [T_1]} m_t \leq 0.1$ **with probability at least** $1 - \delta T_1/(4T)$.    In the sequel, we condition on the good event in Claim I. Let $T_2$ be the first time $t \geq 1$ such that $m_t \leq 0.1$; note that it is possible to have $T_2 > T_1$ or even $T_2 = \infty$. We first show that if $m_t \geq 0.1$, then

$$\mathbb{E}[m_{t+1/2}|\mathcal{F}_t] - m_t \leq -\frac{c_2 \eta_t \sigma_t^2}{d} \tag{15}$$

for some absolute constant $c_2 > 0$. Indeed, for $\sigma_t \leq 0.1, m_t \in [0.1, 0.2]$, and $|X| \leq 1$, we have

$$0 \leq \sqrt{0.99} m_t - \sqrt{0.99} \sigma_t \leq \sqrt{1 - \sigma_t^2}\, m_t + \sigma_t \sqrt{1 - m_t^2} X \leq m_t + \sigma_t < \frac{1}{3}.$$

Since $f'(m) = -1$ for all $m \in [0, 1/3]$ in our construction, Equation (15) follows from Lemma 1.

Next, for every $t \leq \min\{T_2, T_1\}$, we write

$$m_t - m_1 = \sum_{r=1}^{t-1}\left[ \underbrace{\left(\mathbb{E}[m_{r+1/2}|\mathcal{F}_r] - m_r\right)}_{\leq -\frac{c_2 \eta_t \sigma_t^2}{d} \text{ by Equation (15)}} + \underbrace{\left(m_{r+1/2} - \mathbb{E}[m_{r+1/2}|\mathcal{F}_r]\right)}_{=:A_r} + \underbrace{\left(m_{r+1} - m_{r+1/2}\right)}_{\leq 0 \text{ by Lemma 4}} \right].$$

Similar to Claim I, we have $|\sum_{r=1}^{t-1} A_r| \leq 0.05$ with probability at least $1 - \delta/(4T)$. On the other hand, the total drift is

$$\sum_{r=1}^{T_1-1} \left(\mathbb{E}[m_{r+1/2}|\mathcal{F}_r] - m_r\right) \leq -\frac{c_2}{d} \sum_{r=1}^{T_1-1} \eta_t \sigma_t^2 \overset{(a)}{\leq} -\frac{c_2 \log^2(T/\delta)}{cd} \sum_{r=1}^{T_1-1} \eta_t^2 \sigma_t^2 \overset{(b)}{\leq} -\frac{c_1 c_2}{2c},$$

where (a) uses the upper bound of $\eta_t$, and (b) uses the definition of $T_1$. By choosing $c > 0$ small enough, the total drift can be made smaller than $-0.1$, so that if $T_2 > T_1$, then $m_{T_1} \leq m_1 - 0.1 + 0.05 \leq 0.1$, which in turn means that $T_2 \leq T_1$, a contradiction. Therefore, with probability at least $1 - \delta T_1/(4T)$, we have $T_2 \leq T_1$, or equivalently $\min_{t \in [T_1]} m_t \leq 0.1$.

Finally, it is clear that a repeated application of Claim I and II implies Proposition 1: starting from the first time $T_0$ with $m_{T_0} \geq 0.1$, the above claims show that with high probability, future alignment $m_t$ will fall below $0.1$ before it rises above $0.2$. Once $m_t$ falls below $0.1$, we repeat the entire process again and wait for the next time it falls below $0.1$. Since the failure probability at each step of the analysis is at most $\delta/T$, a union bound gives the total failure probability of $\delta$.

## C  AUXILIARY RESULTS

Below we state a self-normalized concentration inequality for martingales (Whitehouse et al., 2023, Theorem 3.1) adapted to our setting.

**Definition 1** (CGF-like function). *A function $\psi : [0, \lambda_{\max}) \to \mathbb{R}_{\geq 0}$ is said to be CGF-like if it is $(a)$ twice continuously-differentiable on its domain, $(b)$ strictly convex, $(c)$ satisfies $\psi(0) = \psi'(0) = 0$, and $(d)$ $\psi''(0) > 0$.*

**Definition 2** (Sub-$\psi$). *Let $\psi : [0, \lambda_{\max}) \to \mathbb{R}_{\geq 0}$ be a CGF-like function. Let $\{S_t\}_{t\geq 0}$ and $\{V_t\}_{t\geq 0}$ be respectively $\mathbb{R}$-valued and $\mathbb{R}_{\geq 0}$-valued processes adapted to some filtration $\{\mathcal{F}_t\}_{t\geq 0}$. We say that $\{S_t, V_t\}_{t\geq 0}$ is sub-$\psi$ if for every $\lambda \in [0, \lambda_{\max})$,*

$$M_t^\lambda := \exp\left(\lambda S_t - \psi(\lambda) V_t\right) \leq L_t^\lambda,$$

*where $\{L_t^\lambda\}_{t\geq 0}$ is a non-negative supermartingale adapted to $\{\mathcal{F}_t\}_{t\geq 0}$.*

The following result is a corollary of (Whitehouse et al., 2023, Theorem 3.1) with the choice $h(k) = (1+k)^2$ for $k \geq 1$.

**Lemma 9** (Self-normalized concentration inequality). *Suppose $\{S_t, V_t\}_{t\geq 0}$ is a real-valued sub-$\psi$ process for $\psi : [0, \lambda_{\max}) \to \mathbb{R}_{\geq 0}$ satisfying*

$$\psi(\lambda) = \frac{\lambda^2}{1 - \lambda/\lambda_{\max}}$$

*on its domain. Let $\delta \in (0, 1)$ denote the error probability. Define the function $\ell : \mathbb{R}_{\geq 0} \to \mathbb{R}_{\geq 0}$ by*

$$\ell_\omega(v) = 2\log\left(1 + \log(v\omega \vee 1)\right) + \log\left(\frac{1}{\delta}\right),$$

*then there exists a universal constant $C > 0$ such that,*

$$\Pr\left(\exists t \geq 1 : S_t \geq C\left(\sqrt{(V_t \vee \omega^{-1}) \ell_\omega(V_t)} + \lambda_{\max}^{-1} \ell_\omega(V_t)\right)\right) \leq \delta.$$

*Proof.* By simple algebra, the convex conjugate $\psi^\star$ of $\psi$ satisfies $(\psi^\star)^{-1}(u) = 2\sqrt{u} + \lambda_{\max}^{-1} u$. The rest follows from (Whitehouse et al., 2023, Theorem 3.1). $\square$

**Lemma 10** (Spherical Stein's Lemma). *Suppose $Z \sim \mathrm{Unif}(\mathbb{S}^{d-1})$ and consider a fixed $\alpha \in \mathbb{R}^d$ and let $X = \langle \alpha, Z \rangle$. For any bounded function $f$,*

$$\mathbb{E}[Xf(X)] = \frac{1}{d-1}\mathbb{E}\left[f'(X)(1 - X^2)\right].$$

*Proof.* The density of $X$ is given by

$$P_d(x) \triangleq \frac{2\left(1-x^2\right)^{\frac{d-1}{2}-1}}{\text{Beta}\left(\frac{1}{2}, \frac{d-1}{2}\right)} \mathbb{I}(|x| \leq 1).$$

Consequently,

$$
\begin{aligned}
\mathbb{E}[Xf(X)] &= \int_{-1}^{1} xf(x) \cdot \frac{2\left(1-x^2\right)^{\frac{d-1}{2}-1}}{\text{Beta}\left(\frac{1}{2}, \frac{k-1}{2}\right)} \mathrm{d}x \\
&\overset{(a)}{=} \frac{2}{d-1} \int_{-1}^{1} f'(x) \cdot \frac{\left(1-x^2\right)^{\frac{d-1}{2}}}{\text{Beta}\left(\frac{1}{2}, \frac{k-1}{2}\right)} \mathrm{d}x \\
&= \frac{1}{d-1} \int_{-1}^{1} f'(x)(1-x^2) \cdot \frac{2\left(1-x^2\right)^{\frac{d-1}{2}-1}}{\text{Beta}\left(\frac{1}{2}, \frac{k-1}{2}\right)} \mathrm{d}x \\
&= \frac{1}{d-1} \mathbb{E}\left[f'(X)(1-X^2)\right],
\end{aligned}
$$

where $(a)$ follows from integration by parts. $\qquad\square$

**Lemma 11.** *Suppose* $X \sim \mathcal{N}(0, I/d)$ *and* $X' \sim \text{Unif}(\mathbb{S}^{d-1})$. *For any fixed* $\alpha \in \mathbb{R}^d$, $\langle \alpha, X \rangle^2$ *dominates* $\langle \alpha, X' \rangle^2$ *in the convex order. Namely, for every convex function* $g : \mathbb{R} \to \mathbb{R}$,

$$\mathbb{E}[g(\langle \alpha, X' \rangle^2)] \leq \mathbb{E}[g(\langle \alpha, X \rangle^2)].$$

*Proof.* Observe that $X$ follows the same distribution as $NX'$, where $N$ and $X'$ are independent, and $N$ is a scaled chi-squared random variable such that $\mathbb{E}[N^2] = 1$. Therefore,

$$
\begin{aligned}
\mathbb{E}[g(\langle \alpha, X \rangle^2)] &= \mathbb{E}[g(N^2 \langle \alpha, X' \rangle^2)] \\
&= \mathbb{E}[\mathbb{E}[g(N^2 \langle \alpha, X' \rangle^2)|X']] \\
&\geq \mathbb{E}[g(\mathbb{E}[N^2] \langle \alpha, X' \rangle^2)] \\
&= \mathbb{E}\left[g(\langle \alpha, X' \rangle^2)\right].
\end{aligned}
$$

$\qquad\square$

