# OpenReview forum: "Interactive Learning of Single-Index Models via Stochastic Gradient Descent"
_ICLR.cc/2026/Conference — ICLR 2026 Poster_

### Official Review · Reviewer_6gnD · 2025-10-14

**Soundness:** 3
**Presentation:** 3
**Contribution:** 2
**Rating:** 4
**Confidence:** 3

**Summary:**

This paper studies a variation of Single Index Models where the features are 'interactive' meaning they depend on the current state of the parameter vector in a specific way and hence the data is not iid. The required sample complexity to learn the unknown parameter vector under SGD is studied. This paper proves sample complexity bounds on estimating the unknown parameter vector to arbitrary precision for a class of link functions which according to other recent works is tight up to poly-logarithmic factors. Additionally for the same set up the authors prove regret bounds.

I think the results shown in this paper are interesting and begin to contribute to filling in the picture of SGD and Single Index Models with different features, however, I think the results are somewhat marginal as there seem to be other very closely related questions regarding sample complexity of these models which are not quite flushed out here. Please see the sections on weaknesses and my questions below. If my concerns can be addressed and I can be convinced that the story is more complete than I currently suspect it to be I would be happy to increase my scores. Currently I think this is a good start to a paper but needs a bit more flushing out to be complete.

**Strengths:**

The paper proves novel results for understanding the training dynamics and sample complexity of SGD for Single Index Models with interactive features. The results are novel and demonstrate the value of SGD in this setting. The paper does a good job at explaining their results and the outlines of the proofs which are often otherwise hidden in the appendix.

**Weaknesses:**

The assumptions feel rather restrictive, particularly monotonicity of the link function. There is discussion of the 'necessity' of the monotonicity assumption, however there is not nearly enough detail to convince me that the assumption is necessary. If the assumption is truly necessary for the results it would be nice to have a rigorous negative result such as a counter example demonstrating that a lack of monotonicity will indeed break the conclusion of the proof.

The relationship of the information exponent in the Gaussian iid feature case is discussed and how the concept does not directly apply to the non-interactive setting. However the information exponent in general relates to the taylor expansion of the population loss and specifically for single index models with iid gaussian data this concept reduces to checking the Hermite coefficients of the link function. It is unsurprising that when the data is not gaussian the Hermite coefficients are no longer important. What would be nice is to consider the original notion of information exponent applied specifically to this problem. For example it is not clear to me that the assumptions (monotonicity and either bounded derivative from below or convexity) do not somehow still just imply information exponent 1 but under a more appropriate definition.

**Questions:**

Do you suspect that there is a similar notion of information exponent for single index models with 'interactive features'? With Gaussian data the sample complexity is of course strongly dependant on the information exponent and can vary widely. Here, only one case is presented: i.e. you can solve the problem with quadratic complexity for the given class of functions. Do you suspect that it is possible to solve the problem for a more general class of link functions given larger sample complexity? Or that outside of the given class you cannot solve the problem? Simulations may be informative as well to answer these questions.

---

> ### Author Response · Authors · 2025-11-19
>
> Thank you for your comments! Please find our responses below:
>
> **1. Monotonicity of link function**
>
> Please refer to point #3 of our general response. This monotonicity assumption is standard in interactive settings; for example, we are closely following the prior work [Rajaraman et al. 2024], where monotonicity is assumed to ensure that reward maximization is aligned with parameter estimation. Once monotonicity is dropped, maximizing the reward is equivalent to finding an action $a$ such that $\langle \theta^\star, a \rangle \approx x^\star$, which may be unrelated to the task of learning $\theta^\star$.
>
> Thanks for your suggestion on including a rigorous statement for our monotonicity discussion in Section 5. We have updated our submission to include a formal statement of this lower bound (Proposition 1 in the revision), showing that there exists a simple non-monotone link function $f$ such that with high probability interactive SGD will fail to escape a basin around the origin as long as the learning rate $\eta_t$ and exploration schedule $\sigma_t$ are not too large. At a high level, this failure is an easy consequence of the non-monotonicity of the population loss in the interactive setting.
>
> **2. Interactive version of information exponent**
>
> As you correctly noted, the conventional notion of an information exponent based on Hermite coefficients is irrelevant in the interactive setting, since the actions are no longer Gaussian. We fully agree; this is precisely why we did not discuss information exponents until Section 5, where the topic appears solely for completeness. Moreover, as summarized in point #1 of our general response, even the information-theoretic sample complexity for nonlinear bandits remains a challenging open problem. The prior work [Rajaraman et al. 2024] has made substantial progress, providing new upper and lower bounds via differential equations, but gaps between these bounds persist in general. Our primary motivation is to show that the best known upper bound can in fact be achieved by a remarkably simple and natural algorithm, namely SGD.
>
> If you are interested in what the statistical complexity is expected to be in the interactive setting, our current conjecture is the expression stated in the last paragraph of Section 5 (Page 10). This has no connection to the information exponent because the interactive setting is fundamentally a different problem. For an explanation of why this conjecture is expected to be tight, as well as the challenges involved in proving the corresponding lower bound, we refer to the discussion in Section 4.4 of [Rajaraman et al. 2024]. If the hypothesis-testing subroutines of [Rajaraman et al. 2024] are combined with SGD, we expect that such a hybrid method could also achieve this upper bound; however, this would no longer be the SGD algorithm itself, and we would lose the beauty of the simple SGD update.
>
> P.s. Somehow we don’t understand your comment “what would be nice is to consider the original notion of information exponent applied specifically to this problem”. Hopefully our comments above have convinced you that the information exponent is simply not the right complexity measure in the interactive setting.
>
> Reference:
>
> [Rajaraman et al. 2024]: Rajaraman, N., Han, Y., Jiao, J., & Ramchandran, K. (2024). Statistical complexity and optimal algorithms for nonlinear ridge bandits. The Annals of Statistics, 52(6), 2557-2582.

---

> > ### Comment · Reviewer_6gnD · 2025-11-19
> >
> > Thank you for the response and revisions. I feel you have addressed my main concern and so I have updated my score.
> >
> > To clarify the last comment “what would be nice is to consider the original notion of information exponent applied specifically to this problem” I just meant that the notion of information exponent applies more broadly than just to Single Index Models with Gaussian features. In that specific case the information exponent reduces to finding the first non-zero Hermite coefficient of the link function, but more generally the information exponent for any given problem requires looking at the first non-zero term in the Taylor expansion of the population loss. What I meant was that perhaps the notion of the information exponent could still apply to this problem, but would no longer translate to Hermite coefficients. Given your counter example, I now suspect that this problem would not fit into even the general framework of the information exponent.

---

> > > ### Author Response · Authors · 2025-11-20
> > >
> > > Thank you for your feedback and appreciation!
> > >
> > > Yes we share the same feeling that the Taylor expansion may no longer give the right measure in the bandit case. When determining the statistical complexity for non-linear bandits, as [Rajaraman et al. 2024] showed, the main task of the burn-in phase seems to be choosing a good action to maximize the information gain at every step.

---

### Official Review · Reviewer_Veb4 · 2025-10-24

**Soundness:** 3
**Presentation:** 4
**Contribution:** 3
**Rating:** 6
**Confidence:** 4

**Summary:**

In this paper, the author study the time/sample complexity of learning single-index models when the learner is allowed
to query a specific, instead of random, point. They show that a simple SGD-type algorithm can achieve the best-known
bounds on this problem, under the assumption that (1) the target function is monotone and has a nonzero derivative
at $0$, or (2) the target function is convex.
At each step, the algorithm queries the reward/target value at a perturbed version of the current weight and update the
weight using the (spherical) SGD. The size of the perturbation controls the exploration-exploitation trade-off:
In the burn-in stage ($1/\sqrt{d}$ to constant correlation) and the exploration stage (constant correlation to $1 - o(1)$
correlation), larger perturbations are used, and in the final exploitation stage (minimizing the regret in the $1 - o(1)$
correlation regime), a small perturbation is used.

**Strengths:**

* Overall, this is a well-written paper and is easy-to-follow. In addition, it is short (19 pages), which is a nice and
  rare thing for a theory paper to have.
* It is somewhat surprising that how being able to choose the position of the query greatly simplifies the analysis and
  improves the bounds (when the label noise is large). They choose the next query position $a_t$ to be a weighted average
  of the current weight $\theta_t$ and a noise that is *orthogonal* to the current weight. This makes the
  $f( \theta_t \cdot a_t )$ part deterministic. Together with the monotonicity/convexity assumption, the analysis
  becomes much cleaner than the usual analysis.

**Weaknesses:**

This is a neat paper that does everything the authors claim to achieve, so I do not think there is any major weakness,
though one could complain that the setting is too easy. Nevertheless, the following are a few complaints I have.

* As someone who is more familiar with IE/Gaussian single-index models, I found the $\tilde{O}(d^2)$ bounds really
  confusing until I realized that the scaling is different, as the non-interactive bounds are $\tilde{O}(d)$. It
  might be better to point this out early on, instead of putting the discussion in the Related Work section and
  Section 5.
* The comparison with the information exponent results is not entirely fair, as the discrepancy comes mainly from the
  label noises $\epsilon_t$. Without the label noise (i.e., we have access to $f(\theta^*, a)$), the IE bounds are
  invariant under rescaling. It seems that being able to choosing the query point does not lead to improvements in
  the no-label-noise setting. Moreover, if the link function is $f(x)=x^{2q}$ for some positive integer $q$ (the convex
  setting), the IE is $2$, so the IE bound is $\tilde{O}(d)$, while Theorem 2(2) depends on $1/f'(1/\sqrt{d})$, which
  can be large when $q$ is large.

**Questions:**

* See the 2nd point of the weakness section. In particular, how do your bounds depend on the size of the label noise?
  Can they recover/improve over the usual IE bounds when there are no label noises or the label noise is small?

---

> ### Author Response · Authors · 2025-11-19
>
> Thank you for your comments, and especially your careful reading on the scaling differences! Here are our responses to your comments:
>
> **1. One could complain that the setting is too easy.**
>
> Well, yes and no. We totally agree that a deterministic inner product $\langle \theta_t, a_t \rangle$ makes our analysis much simpler and cleaner, so our overall argument is short. However, if we set aside the SGD algorithm and consider the underlying non-linear ridge bandit problem studied in [Rajaraman et al. 2024], it is not at all obvious a priori why the problem should be easy! For instance, [Rajaraman et al. 2024] demonstrate that many classical methods, such as UCB or regression-based algorithms, fail to achieve the optimal burn-in cost and require new techniques tailored specifically for that phase. It is therefore striking that the SGD algorithm, despite being so simple and natural, manages to
> - obtain the best known burn-in cost for many link functions, and
> - achieve optimal performance in both the burn-in and learning phases simultaneously.
>
> In this sense, the simplicity of our SGD algorithm is a strength!
>
> **2. The issue with different scaling.**
>
> Thank you for spotting this difference! We were mainly gearing our writing towards the bandit community without sufficiently realizing the cultural differences from the ML community on single-index models; as a result, we only offered a brief discussion below Corollary 1. We have now clarified this issue in point #2 of our general response and made the distinction more explicit in the revision.
>
> **3. Dependence on label noise.**
>
> Thank you for this excellent point. Our results for the interactive setting indeed need a constant level of label noise; once again this noise assumption is very natural in the bandit scenario. We’d also like to point out a subtle difference in the motivation. As explained in point #1 of our general response, even the statistical complexity in the interactive case is highly nontrivial. Therefore, we aim to propose sound algorithms to achieve a near-optimal statistical complexity, and in our paper we **propose** to use SGD despite its simplicity. If there is no label noise, the statistical complexity in the interactive case becomes much smaller and can be addressed by algorithms other than SGD (e.g. the algorithm in [Rajaraman et al. 2024] which requires only $O(d)$ queries with no label noise). By contrast, in the non-interactive case, the literature **chooses** to study SGD because of its practical prevalence and computational efficiency, and its analysis happens to have no to little dependence on the label noise. This is the reason why we focus only on the case with label noise; nevertheless we will add your excellent point to our final paper!
>
> Reference:
>
> [Rajaraman et al. 2024]: Rajaraman, N., Han, Y., Jiao, J., & Ramchandran, K. (2024). Statistical complexity and optimal algorithms for nonlinear ridge bandits. The Annals of Statistics, 52(6), 2557-2582.

---

> > ### Comment · Reviewer_Veb4 · 2025-11-19
> >
> > Thank you for the response. This is a nice paper, and I'll maintain my positive score.

---

> > > ### Author Response · Authors · 2025-11-20
> > >
> > > Thank you for your feedback and appreciation!

---

### Official Review · Reviewer_zDB6 · 2025-11-03

**Soundness:** 2
**Presentation:** 3
**Contribution:** 2
**Rating:** 4
**Confidence:** 3

**Summary:**

This paper considers interactive SGD for single-index bandit problems where the reward is a potentially non-linear function of the dot-product between the current and optimal actions. The paper analyses both a burn-in phase to get a constant dot-product, and a learning phase which starts from a warm-start initialization and achieves a $\tilde{O}(d^2/\varepsilon)$ sample-complexity and $\tilde{O}(d\sqrt{T})$ regret.

**Strengths:**

While there is now a rich literature for learning single-index models in the supervised setting, I believe the literature on online/bandit settings is more sparse. Therefore, the problem that this paper wants to tackle is novel and of importance to the community. Also, the paper is explicit about its assumptions and it is generally easy to read and follow.

**Weaknesses:**

My main concerns are the following:
* In the pure exploration case $\sigma_t = 1$, this algorithm is the same as one-pass SGD studied by Ben Arous et al., 2021. However, the sample complexity seems to be worse. Due to the monotonicity of $f$ (hence information exponent 1), the sample complexity of (pure exploration) SGD would scale linearly with $d$ (at least in the noiseless setting, but I believe it should be able to tolerate $O(1)$ i.i.d. noise as well). However, the bound of Theorem 1 (SGD with warm-start initialization) scales with $O(d^2)$ and that of Corollary 1 can be as large as $O(d^p)$ for $f(x) = x^p$. It might be plausible that to get sub-linear regret, one should ultimately settle for a worse sample complexity, but if that's the case it should be better highlighted.

* The argument of Section 5 only shows that $f$ needs to be monotone around $m \approx 1$, and one can drive $m$ towards $1$ by pure exploration. This accommodates higher order Hermite polynomials, e.g. $H\_2,H\_6,...$. It would be interesting to know the effect of high information exponent in such cases.

**Questions:**

* I think it would be very useful if the schedule of $\sigma_t$ could be presented more explicitly for regret minimization in Corollary 1.

---

> ### Author Response · Authors · 2025-11-19
>
> Thank you for your comments! Here are our responses to your questions:
>
> **1. Sample complexity is worse compared to [Ben Arous et al. 2021]**
>
> This is not the case: our sample complexity for interactive SGD is no worse than the one-pass SGD in [Ben Arous et al. 2021] or any non-interactive algorithms. This is because a different scaling is used in our work, so comparisons must be made with care; please refer to point #2 in our general response for details. Under our scaling, for $f(x) = x^p$ with $p\ge 2$, any non-interactive algorithm (including the one-pass SGD) must incur a sample complexity lower bound $\widetilde{\Omega}(d^{p+1})$ (cf. Theorem 4.1 of [Rajaraman et al. 2024]), but our SGD achieves the optimal sample complexity $\widetilde{\Theta}(d^p)$ (cf. discussions below our Corollary 1).
>
> We choose this different scaling because of the usual convention in the bandit literature. For example, in the simplest example of linear bandits, the actions are usually assumed to lie in an $\ell_2$ ball of radius $O(1)$, rather than $O(\sqrt{d})$. Since the main focus of our work is the interactive/bandit case, we choose this bandit scaling to better parallel with the results in [Rajaraman et al. 2024].
>
> **2. Monotonicity assumption**
>
> Once again, the monotonicity assumption is another major distinction between the non-interactive and interactive cases; we refer to point #3 in our general response for details. Your examples of higher order Hermite polynomials are not monotone, so when using SGD to learn these functions, we essentially need to go back to the non-interactive regime. In the revision, we have also added an explicit counterexample in Proposition 1 showing the failure of SGD without this monotonicity. Although it is still an interesting question to understand how interaction could help for non-monotone function, the main focus of our work is to present a natural algorithm for the setting of [Rajaraman et al. 2024], so by default we adopt their monotonicity assumption.
>
> A small typo in your comment: Sec 5 shows that $f$ needs to be monotone in $[0, \sqrt{1-\sigma^2}]$, therefore monotonicity can be dropped when $\sigma \approx 1$ (not $m\approx 1$ in your comment). However, when $\sigma \approx 1$ the algorithm effectively reverts to a non-interactive setting, eliminating any meaningful benefit from interaction. This is not the regime we aim to study.
>
> References:
>
> [Ben Arous et al. 2021] Ben Arous, G., Gheissari, R., & Jagannath, A. (2021). Online stochastic gradient descent on non-convex losses from high-dimensional inference. Journal of Machine Learning Research, 22(106), 1-51.
>
> [Rajaraman et al. 2024]: Rajaraman, N., Han, Y., Jiao, J., & Ramchandran, K. (2024). Statistical complexity and optimal algorithms for nonlinear ridge bandits. The Annals of Statistics, 52(6), 2557-2582.

---

> ### Comment · Reviewer_zDB6 · 2025-11-19
>
> Thank you for your response. I think that to make a meaningful comparison to the supervised setup, $f$ should be thought of as the (negative) loss rather than the link function. Assuming e.g. a single-index model $z \mapsto \sigma(\langle \theta^\star, z\rangle)$ where $z \sim \mathcal{N}(0,I_d)$, one could define the correlation loss $\ell(z,\theta^\star,a) = -\sigma(\langle \theta^\star, z\rangle)\sigma(\langle a, z\rangle)$, and then define $f$ as
> $$f(\langle \theta^\star, x\rangle) = E\_z[-\ell(z,\theta^\star,a)] + \varepsilon$$
> with the zero mean noise
> $$\varepsilon = -\ell(z, \theta^\star, a) + E\_z[\ell(z,\theta^\star,a)].$$
> Under this formulation, $f(t) = t^p$ corresponds to the case where $\sigma$ is the $p$th Hermite polynomial, and therefore would be an example of the information exponent $p$ case. In particular, $f$ would automatically be monotone on the non-negative real line, which is why Ben Arous et al. assume the initial correlation to be non-negative.
>
> The difference between this algorithm and that of Ben Arous et al. is that they can directly run SGD on the map $a \mapsto f(\langle \theta^*, a\rangle) + \varepsilon$, while here have a different noise structure and don't have access to this information. This would also be a principled way to justify the difference between signal-to-noise rations. With these explanations added to the paper, I'm happy to increase my score.

---

> > ### Author Response · Authors · 2025-11-20
> >
> > Thank you for this very thoughtful comment! However we are a bit confused with the message you'd like to convey. We understand that when $\sigma$ is the $p$-th Hermite polynomial, then $\mathbb{E}_z[\sigma(\langle \theta^\star, z\rangle) \sigma(\langle a, z\rangle)] = \langle \theta^\star, a\rangle^p$. But what's important in the interactive case/bandits is to choose actions $a_t$ depending on the past; our current algorithm chooses $a_t$ by exploring around the current $\theta_t$ and updates $\theta_t$ by running SGD on $\theta\mapsto (f(\langle \theta, a_t\rangle) - r)^2$. Our main result is that this choice of $a_t$ is good (e.g. achieves the best known statistical complexity for $f(t)=t^p$, which is already challenging and only resolved recently by different algorithms). In the supervised case, the distribution of $a_t$ will be fixed, say $a_t\sim \mathsf{Unif}(\mathbb{S}^{d-1})$ or $a_t \sim N(0, I_d/d)$. We're wondering if your discussion still only concerns the supervised/non-interactive setup? Your elaboration would be greatly appreciated!

---

### Author Response · Authors · 2025-11-19
**Overall response**

We thank all reviewers for the valuable comments. We acknowledge that some distinctions may not have been communicated clearly in the current draft, and we have uploaded a revision (with changes in blue) for better clarifications.

Before providing a point-to-point response to each of your comments, we’d like to clarify some of the confusions and misunderstandings you may have about our work.

1. The interactive learning of single-index models is **fundamentally different** from the non-interactive case: in the interactive case, even the statistical (i.e. information-theoretic) complexity becomes challenging. As was witnessed in [Rajaraman et al. 2024], interactive learning in single-index models (or non-linear bandits) exhibits two distinct phases, i.e. the burn-in phase and the learning phase. The main results of [Rajaraman et al. 2024] include:

- Interaction reduces the statistical complexity of the burn-in phase in an adaptive manner, where the optimal learning trajectory is sandwiched between two differential equations (cf. Theorem 1.1 and 1.2 of [Rajaraman et al. 2024]). In contrast, any non-interactive algorithm follows a “linear” trajectory and can be strictly suboptimal (Theorem 4.1 in [Rajaraman et al. 2024]).

- While many classical algorithms (UCB, regression-based methods, etc) achieve optimal performance in the learning phase, they are strictly suboptimal for the burn-in phase. Consequently, new algorithmic ideas are necessary for the burn-in phase.

For a more detailed overview of these findings, we refer the reader to the introduction of [Rajaraman et al. 2024]. In view of these existing results, our new results on SGD for the same task are surprising for several reasons:

+ Statistical: For a subclass of link functions (e.g., convex), SGD matches the best-known sample complexity bounds of [Rajaraman et al. 2024] in the interactive setting. In particular, the SGD trajectory in the burn-in phase is adaptive (taking an integral form). Unlike UCB which is provably suboptimal during the burn-in phase, SGD achieves optimal performance in this phase for many link functions.

+ Algorithmic: Despite its simplicity, our SGD method differs fundamentally from existing interactive burn-in algorithms. As discussed in Section 5, our formulation yields an unbiased estimator of the population gradient, unlike the noisy gradient estimators in the zeroth-order approach of [Huang et al. 2021]. Moreover, the burn-in algorithm in [Rajaraman et al. 2024] relies on a complex hypothesis-testing routine. Finally, SGD achieves optimal performance in both the burn-in and learning phases simultaneously, whereas [Rajaraman et al. 2024] employs distinct algorithms for each phase. This makes SGD an appealing and unified solution for interactive single-index learning.

References:

[Huang et al. 2021] Huang, B., Huang, K., Kakade, S., Lee, J. D., Lei, Q., Wang, R., & Yang, J. (2021). Optimal gradient-based algorithms for non-concave bandit optimization. Advances in Neural Information Processing Systems, 34, 29101-29115.

[Rajaraman et al. 2024]: Rajaraman, N., Han, Y., Jiao, J., & Ramchandran, K. (2024). Statistical complexity and optimal algorithms for nonlinear ridge bandits. The Annals of Statistics, 52(6), 2557-2582.

---

> ### Author Response · Authors · 2025-11-19
> **Overall response (cont'd)**
>
> 2. Our work uses a **different scaling** convention from [Ben Arous et al. 2021], so comparisons between results must be made with care (as also noted by Reviewer Veb4):
> - Why the scalings differ: In the non-interactive case (such as [Ben Arous et al. 2021]), the actions/data satisfy $x_t \sim N(0, I_d)$, which implies $\|x_t\| \asymp \sqrt{d}$. In the interactive setting, following standard conventions in linear bandits, our actions $a_t$ lie in the unit ball, so $\|a_t\| \le 1$. Therefore, the SNR in the interactive case is significantly smaller than the SNR in the non-interactive case.
> - How the different scalings affect the sample complexity: for $f(x) = x^p$ with $p\ge 2$, the statistical complexity in the non-interactive case is $\widetilde{\Theta}(d)$ under the scaling of [Ben Arous et al. 2021]. Under our scaling, however, this becomes $\widetilde{\Theta}(d^{p+1})$. For instance, Theorem 4.1 of [Rajaraman et al. 2024] proves a lower bound $\widetilde{\Omega}(d^{p+1})$ any non-interactive algorithm. In the interactive setting, by contrast, Example 1 of [Rajaraman et al. 2024] shows that the optimal sample complexity is $\widetilde{\Theta}(d^p)$, and our SGD achieves the same bound. This example is also discussed in the paper, below Corollary 1.
>
> Therefore, our results are never worse than the non-interactive guarantees. Moreover, interactive SGD strictly outperforms all possible non-interactive algorithms, including non-interactive SGD. We have uploaded a revision with additional remarks in Section 1 and Section 5 on this different scaling.
>
> References:
>
> [Ben Arous et al. 2021] Ben Arous, G., Gheissari, R., & Jagannath, A. (2021). Online stochastic gradient descent on non-convex losses from high-dimensional inference. Journal of Machine Learning Research, 22(106), 1-51.
>
> [Rajaraman et al. 2024]: Rajaraman, N., Han, Y., Jiao, J., & Ramchandran, K. (2024). Statistical complexity and optimal algorithms for nonlinear ridge bandits. The Annals of Statistics, 52(6), 2557-2582.

---

> ### Author Response · Authors · 2025-11-19
> **Overall response (cont'd)**
>
> 3. **Monotonicity assumption** on the link function $f$: Once again, the monotonicity assumption is another major distinction between the non-interactive and interactive cases. In the non-interactive case, the monotonicity assumption is not needed, and SGD learning is characterized solely by the information exponent. In the interactive case, however, if we aim to capture the interactive nature and improve over the non-interactive learning performance, the monotonicity assumption turns out to be somewhat necessary. We provide two justifications:
> - The prior work [Rajaraman et al. 2024], which we aim to parallel, adopts the same assumption, and it is essentially the only structural assumption made there. In their bandit setting, monotonicity ensures that reward maximization is aligned with parameter estimation: improving the alignment between $a_t$ and $\theta^\star$ directly increases the learner’s reward, reflecting the interactive nature of gradually improving $\langle \theta^\star, a_t \rangle$. Once the monotonicity is dropped, for example when $f(x)$ is maximized at an interior point $x^\star\in (0,1)$, the situation changes. Instead of seeking nearly perfect alignment $\langle \theta^\star, \widehat{\theta} \rangle \approx 1$, the learner must search for an action $a$ satisfying $\langle \theta^\star, a \rangle \approx x^\star$, an unnatural target.
> - In the context of our interactive SGD, monotonicity plays an additional essential role, since the population loss must decrease with the alignment $\langle \theta^\star, \theta_t \rangle$ for SGD to succeed at the population level. This property holds in the non-interactive setting of [Ben Arous et al. 2021] (stated as Assumption A, and verified in Proposition 2.1). For our interactive setting, Section 5 shows that achieving monotonicity of the population loss requires the function $f$ to be monotone on $[0, \sqrt{1-\sigma^2}]$ as well. Therefore, without monotonicity, SGD dynamics will become trapped at a local maximizer of $f$ and fail to make further progress unless (1) $\sigma_t$ is very close to 1; or (2) a large learning rate is used to escape this local maximizer. In case (1), when $\sigma_t$ is very close to 1, the algorithm essentially collapses to a non-interactive setting, eliminating the benefits of interaction. In case (2), analyzing escape dynamics under a large learning rate tends to be technical and highly problem-specific. Neither of them is the regime we aim to study.
>
> For the above reasons, we regard the monotonicity assumption as fundamental in the interactive case. While it may be possible to relax or remove this assumption, doing so lies beyond the scope of the present work. In the revision we have expanded these points in Section 1 and Section 5, and **added an explicit counterexample** in Proposition 1 showing the failure of SGD without this monotonicity.
>
> References:
>
> [Ben Arous et al. 2021] Ben Arous, G., Gheissari, R., & Jagannath, A. (2021). Online stochastic gradient descent on non-convex losses from high-dimensional inference. Journal of Machine Learning Research, 22(106), 1-51.
>
> [Rajaraman et al. 2024]: Rajaraman, N., Han, Y., Jiao, J., & Ramchandran, K. (2024). Statistical complexity and optimal algorithms for nonlinear ridge bandits. The Annals of Statistics, 52(6), 2557-2582.

---

### Author Response · Authors · 2025-12-01
**Author final remark**

Dear AC,

Thank you for handling our paper. The OpenReview incident was indeed unfortunate, and we are grateful to your additional effort during the hard time. Since the author response period is ending soon, we'd like to summarize some of the confusions and misunderstandings in the original reviews, and how they were clarified in the response.

1. **Different scaling**: Reviewers zDB6 and 6gnD were confused about why our SGD sample complexity in the interactive case appears worse than the one given by information exponent. In fact, as also noticed by Reviewer Veb4, this is not the case. Adopting the convention in the bandit literature, we assume that the actions are in the unit ball (of norm $1$); by contrast, in the non-interactive study of single index models, the features are often $N(0, I_d)$ (of norm $\sqrt{d}$) so have a larger SNR. As a result, our results are never worse than the non-interactive guarantees under the same scaling. For instance, for the link $f(x) = x^p$ with $p\ge 2$, the statistical complexity in the non-interactive case is $\widetilde{\Theta}(d^{p+1})$ under our bandit scaling (achieved by non-interactive SGD, with lower bound in [Rajaraman et al. 2024, Theorem 4.1]). In the interactive setting, the optimal sample complexity is $\widetilde{\Theta}(d^p)$ ([Rajaraman et al. 2024, Example 1]), and our SGD achieves the same bound. We have added several remarks to the revised manuscript.

2. **Monotonicity assumption**: Reviewers zDB6 and 6gnD were also confused about the monotonicity assumption in our work, an assumption which is not required in the non-interactive case. In fact, in both non-interactive and interactive cases, the population loss must decrease with the alignment $\langle \theta^\star, \theta_t \rangle$ for SGD to succeed at the population level. In the non-interactive case, it was stated as Assumption A and verified in Proposition 2.1 in [Ben Arous et al. 2021]. In the interactive case, we showed in Section 5 that monotonicity of the link is necessary unless we essentially go back to the non-interactive scenario. In the revision, we also added a new result (Proposition 1) that provably shows this necessity via a concrete example.

3. **Difficulty of interactive learning**: Although our work draws a clear connection with the rich line of study of single-index models (in which all reviewers have great expertise), we'd like to remark that the interactive case (or the nonlinear ridge bandit) is fundamentally different. For instance, even its statistical complexity is challenging (and only partially addressed in [Rajaraman et al. 2024]). Therefore, instead of **choosing** to study the learning dynamics of SGD, we are also making the algorithmic contributions of **proposing** to use SGD for nonlinear ridge bandits and proving that it is natural, simple, unified (works for both burn-in and learning phases), and sound (matches best known upper bounds of [Rajaraman et al. 2024]).

For more detailed discussions, please feel free to read our overall response.

All of our discussions with the reviewers happened one week before the incident, and the reviewers were happy with our response. Before the scores reverted back, Reviewer zDB6 raised the score to 6, Reviewer Veb4 maintained the score (6), and Reviewer 6gnD raised the score to 8. We hope you could take these factors into consideration in your final recommendation, and we are happy to help in any way that facilitates your decision process (if author-AC communications are still allowed by the system).

Best,

Authors

References:

[Ben Arous et al. 2021] Ben Arous, G., Gheissari, R., & Jagannath, A. (2021). Online stochastic gradient descent on non-convex losses from high-dimensional inference. Journal of Machine Learning Research, 22(106), 1-51.

[Rajaraman et al. 2024]: Rajaraman, N., Han, Y., Jiao, J., & Ramchandran, K. (2024). Statistical complexity and optimal algorithms for nonlinear ridge bandits. The Annals of Statistics, 52(6), 2557-2582.

---

### Meta-Review · Area_Chair_rxa3 · 2025-12-14

**Summary:**

Reviewers have indicated the following concerns:
- SGD sample complexity in the interactive case appears worse than the one given by information exponent
- The requirement of monotonicity assumption for the link function, which is not required in the non-interactive case
- The comparison with the information exponent results is not entirely fair
- Original notion of information exponent is required for non-Gaussian data

**Reviewer Concerns:**

All the reviewers participate actively in the discussion. Throughout the discussions I can see all the concerns are addressed:
- SGD sample complexity in the interactive case appears worse than the one given by information exponent (the authors have clarified the difference between the non-interactive case and the interactive case)
- The requirement of monotonicity assumption for the link function, which is not required in the non-interactive case (the authors have added a new result to show the necessity of monotonicity assumption in the interactive case)
- The comparison with the information exponent results is not entirely fair (the authors indicated that statistical complexity in the interactive case is highly nontrivial, and noise assumption is very natural in the bandit scenario)
- Original notion of information exponent is required for non-Gaussian data (the authors have provided counter example to show this problem would not fit into even the general framework of the information exponent)

**Reviewer Scores:**

Reviewer zDB6 indicated explicitly that he/she was happy to increase the score

Reviewer Veb4 said that he/she would maintain the positive score

Reviewer 6gnD said that his/her comments were addressed well and would like to increase the score

---

### Decision · Program_Chairs · 2026-01-26

Accept (Poster)